# Quantum equilibrium propagation for efficient training of quantum systems based on Onsager reciprocity

**Clara C. Wanjura** [1] ✉ **& Florian Marquardt** [1,2]

The widespread adoption of machine learning and artificial intelligence in all branches of science and technology creates a need for energy-efficient, alternative hardware. While such neuromorphic systems have been demonstrated in a wide range of platforms, it remains an open challenge to find efficient and general physics-based training approaches. Equilibrium propagation (EP), the most widely studied approach, has been introduced for classical energy-based models relaxing to an equilibrium. Here, we show a direct connection between EP and Onsager reciprocity and exploit this to derive a quantum version of EP. For an arbitrary quantum system, this can now be used to extract training gradients with respect to all tuneable parameters via a single linear response experiment. We illustrate this new concept in examples in which the input or the task is of quantum-mechanical nature, e.g., the recognition of many-body ground states, phase discovery, sensing, and phase boundary exploration. Quantum EP may be used to solve challenges such as quantum phase discovery for Hamiltonians which are classically hard to simulate or even partially unknown. Our scheme is relevant for a variety of quantum simulation platforms such as ion chains, superconducting circuits, Rydberg atom tweezer arrays and ultracold atoms in optical lattices.

As deep learning and artificial intelligence are adopted in all fields of science and technology, the increasing complexity of neural networks has led to an exponential increase in energy consumption and training costs. This has created a need for more efficient alternatives, sparking the rapidly developing field of neuromorphic computing[1], which explores a variety of different platforms[2,3] to design physical, analogue neural networks.

Existing training strategies for neuromorphic platforms include in-silico training, requiring a faithful digital model of the system, and physics-aware backpropagation[4], combining physical inference with a simulated backward pass which relaxes these constraints. However, it is a central question whether not only inference but also training can exploit the physical dynamics[5], making full use of the energy efficiency of neuromorphic systems. For example, feedback-based parameter shifting does not require any simulation but scales unfavourably with

the network size[6]. Moving towards physical implementations of efficient backpropagation (the central technique for training artificial neural networks), strategies for specific types of non-linearities have been developed[7–9], as well as approaches performing backpropagation only on the linear components[10,11]. Another novel recent approach enables 'forward-forward' type gradient calculation in systems which perform sequential information processing[12]. Furthermore, efficient measurements of gradients via scattering experiments can be performed in optical systems that employ a framework recently developed to produce nonlinear computation with linear wave setups[13]— such nonlinear processing was also recently demonstrated in refs. 12,14,15.

General approaches for physical backpropagation so far only exist in two classes of physical systems: Hamiltonian Echo Backpropagation[16], which applies to essentially lossless systems in which a time-reversal

[1]Max Planck Institute for the Science of Light, Erlangen, Germany. [2]Department of Physics, University of Erlangen-Nuremberg, Erlangen, Germany.
✉e-mail: clara.wanjura@mpl.mpg.de

operation can be implemented, and equilibrium propagation (EP)[17,18], which applies to energy-based, equilibrating systems.

EP is currently the most intensively studied physics-based training procedure for neuromorphic systems. It stands in the tradition of contrastive learning approaches, comparing measurements obtained from two different equilibria and using feedback to update parameters. Concretely, EP consists of two phases: the free and the nudged phase. In the free phase, the input is fixed and the system relaxes into its equilibrium state. In the nudged phase, the output is 'nudged' closer towards the target output and the system evolves to a new equilibrium. Comparing the system state in the free and the nudged phase, respectively, one obtains the necessary gradients which are then used to update parameters.

Since its introduction in 2017, EP has been investigated thoroughly[19–24] and a variant, coupled learning, has been developed[25,26]. In particular, EP was proposed for training nonlinear resistor circuits[19], systems of coupled phase oscillators[24], was further adapted to spiking networks[20,27], to implement continual parameter updates[21] and a dynamical version was developed[22]. Experimentally, EP has been applied to train electronic systems[28,29], elastic networks[30] and even a memristor crossbar array[31]. Furthermore, a classical Ising model has been trained using a quantum annealer to efficiently reach equilibrium[32].

Given the elegance and wide-ranging impact of the EP approach, it is a natural question to ask whether it can be extended to quantum systems to train a fully quantum Hamiltonian via a nudging procedure similar to classical EP. In this work, we will show that there is a direct connection between EP and Onsager reciprocity and exploit it to derive a quantum version of EP (QEP). This new approach can be used to train efficiently arbitrary quantum systems, including specifically the highly tuneable quantum many-body systems realized nowadays in quantum simulators[33].

We illustrate this new concept with supervised and unsupervised learning examples of a quantum-mechanical nature. Specifically, we investigate the recognition of quantum many-body phases and introduce as new concepts the exploration of phase diagrams with quantum simulator platforms via efficient gradient descent optimization enabled by QEP as well as the optimisation of sensitivity (e.g. for sensing applications) and phase boundary exploration. QEP is applicable to systems which are hard or impossible to simulate classically and can be employed even in settings in which the Hamiltonian is only partially known or partially accessible.

In terms of the general question of using quantum devices for learning tasks, the area of quantum machine learning[34,35] by now has a long history. There have been some ideas of how to learn to reproduce quantum states by adapting tuneable parameters ('quantum Boltzmann machines', see ref. [35]). The major research efforts in this domain are, however, spent on variational quantum circuits, which require fully controllable digital quantum computing platforms for implementation (possibly even fault-tolerant), imposing resource demands that go significantly beyond what we are going to assume here. We provide an overview of digital and analogue neuromorphic computing approaches in the classical and quantum domain in Fig. 1c.

## Results

### Onsager reciprocity as the basis for quantum equilibrium propagation

Consider a parameterized Hamiltonian

$$\hat{H}(\lambda) = \sum_j \lambda_j \hat{A}_j \tag{1}$$

and its ground state $|\Psi(\lambda)\rangle$. A small static force coupling to $\hat{A}_j$ (entering as a term $\delta\lambda_j \hat{A}_j$ inside $\hat{H}$) will produce a linear response in the expectation value $\langle \hat{A}_\ell \rangle$, given by

$$\chi_{\ell j} = \frac{\partial}{\partial \lambda_j} \langle \Psi(\lambda) | \hat{A}_\ell | \Psi(\lambda) \rangle. \tag{2}$$

Onsager reciprocity guarantees the symmetry of the susceptibility $\chi_{j\ell} = \chi_{\ell j}$, i.e., the same effect will be produced by a force acting on $\hat{A}_\ell$ influencing the expectation value $\langle \hat{A}_j \rangle$, see Fig. 1, i.e.,

$$\frac{\partial}{\partial \lambda_j} \langle \hat{A}_\ell \rangle = \frac{\partial}{\partial \lambda_\ell} \langle \hat{A}_j \rangle. \tag{3}$$

Onsager reciprocity[36] can be derived in many ways, also for the quantum case[37]. However, the most elementary approach for static response situations such as the one considered here and applied to the ground state in particular uses first-order perturbation theory for the deformation of the ground state $\partial_{\lambda_j} |\Psi(\lambda)\rangle = (E(\lambda) - \hat{H}(\lambda))^{-1} (\hat{A}_j - \langle \hat{A}_j \rangle) |\Psi(\lambda)\rangle$. Expression (3) also holds for thermal states for which the expectation values above are replaced by $\langle \hat{A}_j \rangle = \text{Tr}(\hat{\rho}\hat{A}_j)$, so the following results apply for arbitrary-temperature quantum equilibrium states. In the case of ground-state degeneracy, expression (3) also holds with $\hat{\rho} = \sum_j |\Psi_j(\lambda)\rangle \langle \Psi_j(\lambda)|$ in which $j$ sums over all degenerate states; this is approximately equivalent to a thermal state at small, but non-vanishing, temperature. Classical Onsager reciprocity is equivalent to what has been termed the 'Fundamental Lemma' for classical EP[17,18].

We will now show that this well-known result (3) gives us access to a general version of equilibrium propagation for quantum systems. To understand that, we first consider supervised learning. Let us assume that the set of operators $\hat{A}_j$ is split into degrees of freedom relating to the input $j \in \mathcal{S}_{\text{in}}$, trainable variables $j \in \mathcal{S}_{\text{train}}$, and the output $j \in \mathcal{S}_{\text{out}}$. Accordingly, the set of parameters $\lambda$ is split into the input $x$ containing the parameters corresponding to the input $\lambda_j = x_j$, the set of training parameters $\theta$ with $\lambda_j = \theta_j$ and the set of couplings to the output observables $\nu$ with $\lambda_j = \nu_j$. Hence, a general QEP Hamiltonian is of the form $\hat{H}(x, \theta, \nu)$ in which $\nu = 0$ during inference. For any given training sample, the input $x$ is fixed by applying a field to all input degrees of freedom (we may write $\lambda_k = x_k$ for $k \in \mathcal{S}_{\text{in}}$, with $x$ representing the input vector). The output is then read off as the expectation values $y_\ell = \langle \hat{A}_\ell \rangle$ in the operators $\ell \in \mathcal{S}_{\text{out}}$. Note that $\hat{A}_\ell$ can be chosen as a projector, in which case $y_\ell$ becomes the probability of obtaining a particular outcome in a measurement; this is useful for classification tasks.

In supervised learning, we are interested in adjusting the trainable parameters $\theta$ in order to 'nudge' the output closer to the desired target output $y^{\text{target}}(x)$, for any given input $x$. More generally, we aim to reduce the loss function $\mathcal{L}(y, y^{\text{target}})$, or rather its average $\bar{\mathcal{L}}$ over many training samples $(x, y^{\text{target}}(x))$, via gradient descent: $\delta\theta = -\eta \partial\bar{\mathcal{L}}/\partial\theta_j$, in which $\eta$ is the learning rate. To do this, we need to obtain the influence of a change in $\theta_j$ on any of the outputs $y_\ell$. For a given fixed training sample, this is just the susceptibility $\chi_{\ell j}(\lambda) = \partial\langle \hat{A}_\ell \rangle / \partial\theta_j$. Evaluating $\chi_{\ell j}$ for all possible trainable parameters $j$, see Fig. 1a, scales unfavourably, requiring a number of different experiments that scale linearly in the number $N$ of these parameters. Accessing the training gradient in this way amounts to the parameter-shift method which is always applicable in any neuromorphic platform but should generally be avoided whenever possible due to this unfavourable scaling. However, Onsager reciprocity, Eq. (3), tells us that we can also access the susceptibility $\chi_{\ell j}$ by performing an alternative, much more efficient experiment that instead reveals $\chi_{j\ell}$: apply a small force $\nu$ acting on the outputs and observe its influence on the expectation values of the degrees of freedom $\langle \hat{A}_k \rangle$ connected to the trainable parameters: $\chi_{j\ell}(\lambda) = \partial\langle \hat{A}_j \rangle / \partial\nu_\ell$, see Fig. 1b. The required measurements of $\langle \hat{A}_j \rangle$ (for any $j$) under application of a given force component $\nu_\ell$ (at fixed $\ell$) can be performed in parallel, in a single experiment. Thus, this approach would already

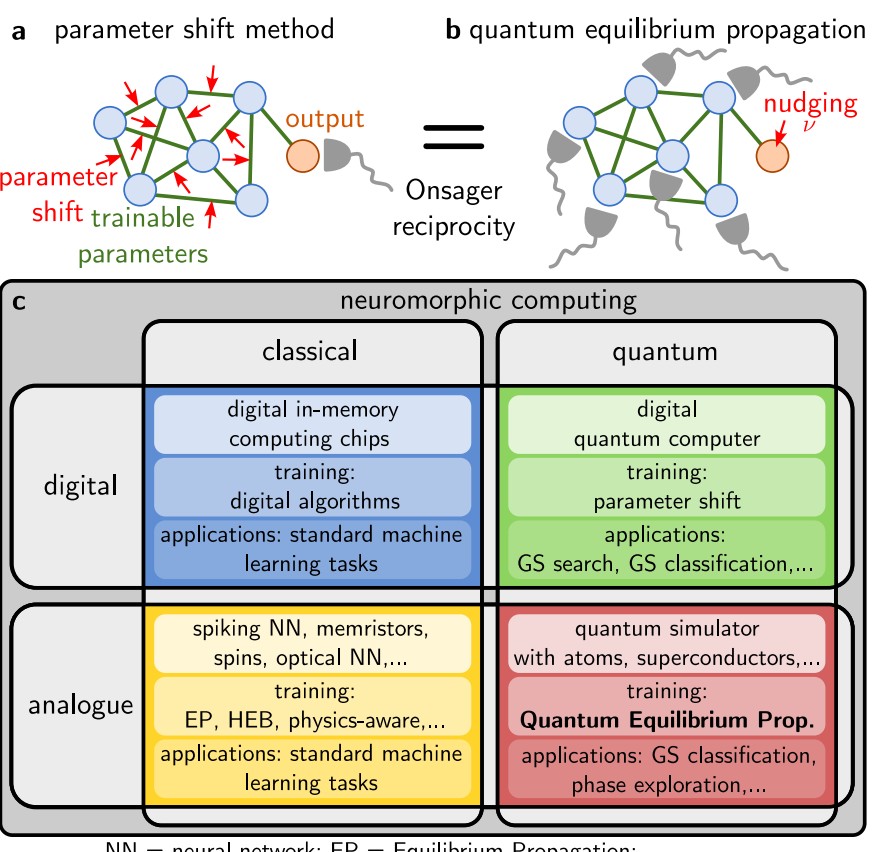

**a** parameter shift method

**b** quantum equilibrium propagation

Onsager reciprocity

**c** neuromorphic computing

| | classical | quantum |
|---|---|---|
| **digital** | digital in-memory computing chips / training: digital algorithms / applications: standard machine learning tasks | digital quantum computer / training: parameter shift / applications: GS search, GS classification,… |
| **analogue** | spiking NN, memristors, spins, optical NN,… / training: EP, HEB, physics-aware,… / applications: standard machine learning tasks | quantum simulator with atoms, superconductors,… / training: **Quantum Equilibrium Prop.** / applications: GS classification, phase exploration,… |

NN = neural network; EP = Equilibrium Propagation;
HEB = Hamiltonian Echo Backpropagation; GS = ground state

**Fig. 1 | The concept of quantum equilibrium propagation.** The goal is to efficiently and in a physical way obtain the gradient of some loss function (depending on expectation values measured at the 'output' degrees of freedom of a quantum system) with respect to tuneable parameters. **a** Rather than shifting $N$ parameters separately and measuring the output expectation value for each shift (parameter-shift method), Onsager reciprocity dictates that the same information can be extracted by (**b**) shifting, i.e. nudging, only the parameters coupling to the output observables and (in a single go) measuring the response of all $N$ operators coupled to the training parameters. This procedure, termed quantum equilibrium propagation, is more efficient as it requires only a single response experiment (or at most a small number of order 1, when some non-commuting observables have to be measured) independent of the system size, whereas the parameter shift method requires a number of experiments scaling linearly with the number of parameters. **c** Overview inspired by ref. [60] of digital and analogue neuromorphic computing schemes and platforms in the classical and quantum regime. Quantum equilibrium propagation can be applied to analogue quantum platforms such as quantum simulators with atoms and superconductors.

seem to require a number of experiments that only scales with the number of outputs $N_{out}$ (number of choices $\ell$), which is typically much smaller than the number of trainable parameters. This already would offer a substantial speedup compared with the naive parameter shift method. However, by evaluating explicitly the desired gradient of the loss function, it becomes apparent that only a single experiment is in fact needed: in this experiment, a force vector $\varepsilon = \partial\mathcal{L}/\partial y$, the so-called error signal, is applied to the output degrees of freedom.

In this way, Onsager reciprocity teaches us how to translate the classical equilibrium propagation approach to quantum devices. Since Onsager reciprocity holds generally in equilibrium systems (SI), quantum Hamiltonians of arbitrary structure can be considered.

**Quantum equilibrium propagation procedure**

We now explicitly summarize the QEP procedure. For clarity, we will from now on denote the output observables by $\hat{O}_\ell$. For a gradient-descent parameter update, we need to compute the derivative of the loss function,

$$\frac{\partial}{\partial\theta_j}\mathcal{L}(y, y^{target}(x)) = \sum_\ell \frac{\partial\mathcal{L}}{\partial y_\ell}\frac{\partial y_\ell}{\partial\theta_j} = \varepsilon\frac{\partial y}{\partial\theta_j} \quad (4)$$

in which the error signal vector has components $\varepsilon_\ell = \partial\mathcal{L}/\partial y_\ell$. For a mean-square-error loss function, we would have $\varepsilon = 2(y - y^{target}(x))$.

The QEP procedure for supervised learning can be summarized as follows. (i) Free phase: The nudging forces are off ($v_j = 0$ for all $j$) and the output expectation values $y_\ell = \langle\hat{O}_\ell\rangle$ as well as the expectation values of all operators associated with trainable parameters $\hat{A}_j$ are measured in the ground state of the Hamiltonian $\hat{H}(x, \theta, 0)$. (ii) Nudged phase: We compute the error signal $\varepsilon$ and use it to nudge the Hamiltonian $\hat{H}(x, \theta, \nu = \beta\varepsilon)$ by switching on the couplings to the output observables, adding a term $\sum_\ell \nu_\ell\hat{A}_\ell$ to the Hamiltonian. The couplings are given by the vector $\nu = \beta\varepsilon$, in which $\beta$ is a small parameter (keeping with the notation for classical EP[17]; this is unrelated to the inverse temperature). We again measure the expectation values of all observables $\hat{A}_j$. (iii) Parameter update: Using Onsager reciprocity, Eq. (3), $\partial\langle\hat{O}_\ell\rangle/\partial\theta_j = \partial\langle\hat{A}_j\rangle/\partial\nu_\ell$, we can approximate the gradient $\partial\langle\hat{O}_\ell\rangle/\partial\theta_j$ and hence arrive at

$$\frac{\partial}{\partial\theta_j}\mathcal{L}(y, y^{target}(x)) \approx \frac{\langle\hat{A}_j\rangle|_{\nu=\beta\varepsilon} - \langle\hat{A}_j\rangle|_{\nu=0}}{\beta}. \quad (5)$$

In a similar spirit as for classical Equilibrium Propagation, one may consider variants which combine positive and negative nudging[38], i.e., approximate the gradient using $(\langle\hat{A}_j\rangle|_{\nu=\beta\varepsilon} - \langle\hat{A}_j\rangle)|_{\nu=-\beta\varepsilon}/(2\beta)$, which empirically performs better for finite nudging.

## Experimental requirements

We now discuss the most important practical considerations for implementing quantum equilibrium propagation (QEP) in any experimental platform.

Above all, the platform needs to have tuneable couplings $\theta$, whose number preferably should be easy to scale up with growing system size. Such tuneable couplings have been developed for many quantum simulators[33] and quantum computing platforms by now. Examples include: (i) ion chains, for which spin-spin couplings can be mediated and engineered via the vibrational modes of the chain, employing suitable Raman transitions., e.g.[39]; (ii) superconducting-qubit arrays, as employed for quantum computing, with current-tuneable couplers between neighbouring qubits[40] and tuneable qubit energies; (iii) neutral-atom Rydberg atom tweezer arrays providing tuneable spin-spin couplings[41]; (iv) strongly interacting atoms in optical lattices with spatially engineered potential energies, hopping amplitudes and interactions, e.g. based on holographic potential shaping[42]. Other platforms, e.g. in optomechanical arrays, coupled microwave cavities, or coupled laser arrays, also demonstrate interesting tuneable coupling schemes, but they often operate out of equilibrium and are therefore not directly suitable for QEP, unless one can map them back to an equilibrium situation.

Beyond this primary requirement of a scalable number of tuneable couplings, QEP platforms also need ways to apply the output forces $v$. This demands local fields, e.g. effective magnetic fields or qubit detunings, easily available in most platforms that are flexible enough to support tuneable couplings. In addition, the expectation values of both the output operators and of the coupling operators connected to trainable degrees of freedom should be measurable. Regarding the couplings, we note that the expectation values of interaction terms of the form $\hat{X}_j \hat{X}_k$ or similar can easily be measured even by observing the spin operators $\hat{X}_j$ individually (and multiplying outcomes). A Heisenberg-type coupling operator $\sum_{\alpha=x,y,z} \hat{\alpha}_j \hat{\alpha}_k$ would need three separate measurements, for the x,y,z components, performed in independent shots of the experiment, eventually obtaining the expectation value of the complete operator. Alternatively, one could carry out a collective (two-qubit) measurement. The latter is typically performed via an ancilla, as demonstrated for syndrome extraction in quantum error correction schemes, and therefore requires more experimental effort. We note that the statistical nature of quantum physics in any case requires many runs of the experiment to measure the expectation values of output variables (needed for inference) and of coupling operators (needed for training). Each of these runs involves an equilibration step.

Finally, QEP requires efficient experimental means to approach the equilibrium state, i.e., for the zero-temperature limit, the ground state $|\Psi(\lambda)\rangle$ of the Hamiltonian. Before discussing physical equilibration, we note that mathematically/computationally the task of ground state search for arbitrary Hamiltonians can become hard both for the classical case (NP-hard for general local spin glass Hamiltonians) and for the quantum case (QMA-hard, i.e., hard even for quantum computers in some instances). At the same time, recently, it has been shown[43] that there exist local quantum Hamiltonians for which it would be classically hard to obtain the ground state that nevertheless can be reached via thermalization. Coming back to physical equilibration dynamics, the situation in quantum EP is analogous to classical EP, in which equilibration needs to be studied on a case-by-case basis. Fortunately, efficient experimental ground-state preparation of complex quantum many-body Hamiltonians is one of the most intensively researched questions in quantum simulation and quantum computing, mirroring analogous progress in classical equilibration[44]. For the purposes of QEP, we distinguish between two options: (i) hybrid approaches, for which an external digital computer is employed during equilibration, and (ii) purely autonomous schemes. Hybrid approaches could rely on variational quantum eigensolvers[45], for which an ansatz

quantum circuit with continuously parametrized unitary gates is performed, the expectation value of the Hamiltonian is measured, and a classical optimization is performed to adapt parameters of the circuit. At first glance, the use of a classical optimizer to find the quantum ground state in this way might seem to contradict the basic premise of QEP or neuromorphic computing in general, i.e., using a physical system to do information processing. However, if the problem setting makes efficient use of the resulting quantum many-body ground state, even such a hybrid approach may still yield an advantage over an entirely classical digital device, in the same way that variational quantum eigensolvers are thought to be beneficial under the right circumstances as compared to numerical ground state search by classical algorithms.

Purely autonomous equilibration schemes get rid of any feedback loop. In principle, coupling to a cold environment can be sufficient, but recently, there has been active research into speeding up equilibration. The techniques put forward often rely in one way or another on variations of quantum reservoir engineering[46], which introduce dissipation deliberately, and which have, e.g., been used to stabilize quantum many-body states in superconducting circuits[47]. This can happen in the form of suitable continuous driving (inspired by laser cooling), e.g. ref. [48], or else in the form of quantum circuits that introduce gates coupling the quantum many-body system to ancilla qubits which then may be periodically reset ('digital quantum cooling', e.g. ref. [49]) or other schemes to reduce entropy, such as via suitable measurements[50].

## Supervised learning: phase recognition in a quantum many-body system

We now move to illustrative applications of QEP. The input considered above is classical, and in the simplest setting QEP could be used to train a quantum device to perform an essentially classical machine learning task. However, we now turn to an important class of applications for which a QEP-trained system can effectively receive input that is quantum instead of classical.

In general, this setting can be realized by starting from a Hamiltonian $\hat{H}_0(x)$, whose quantum ground state we want to analyze with the help of a QEP setup. By tuning the classical parameters $x$ (typically a few), we are able to realize different phases with different ground states, e.g., sweeping through some phase diagram in the case of a quantum many-body system. One task could consist in predicting, for any given $x$, the distinct quantum phase that the system assumes, possibly after seeing a few labelled training examples at a few parameter locations $x$. Another task could consist of approximating the entanglement entropy between some subsystems of $\hat{H}_0$ or predicting any other quantity of interest that can be derived in principle by inspecting the ground state but may be hard to extract directly by simple measurements.

Any of these tasks can be addressed via QEP in the following way. We couple the system of interest, described by $\hat{H}_0(x)$, to a trainable physical sensor, described by $\hat{H}_{sens}(\theta)$, Fig. 2a. Overall, coming back to our previous definitions, we thus have the full QEP Hamiltonian $\hat{H}(x,\theta) = \hat{H}_0(x) + \hat{V}(\theta) + \hat{H}_{sens}(\theta)$, in which we assume that the couplings between the two systems reside inside $\hat{V}$ (see Fig. 2).

Ideally, the couplings inside the system of interest, $\hat{H}_0$, should be stronger than the couplings to the sensor and within that sensor. This will ensure that the system of interest is only weakly perturbed, while the recognition model can still react strongly to the features of the ground state $|\psi_0\rangle$ of $\hat{H}_0$. To ensure that the system-sensor couplings remain small in our numerical experiment (below), we enforce a soft cutoff on the couplings during the training (SI).

Distinguishing quantum phases can serve as an important application, and it has been considered before in the context of quantum machine learning based on gate-based quantum computers, for which a unitary circuit acts on a given ground state encoded in a multi-qubit register[51–53]. In contrast to that, QEP relies on equilibration and

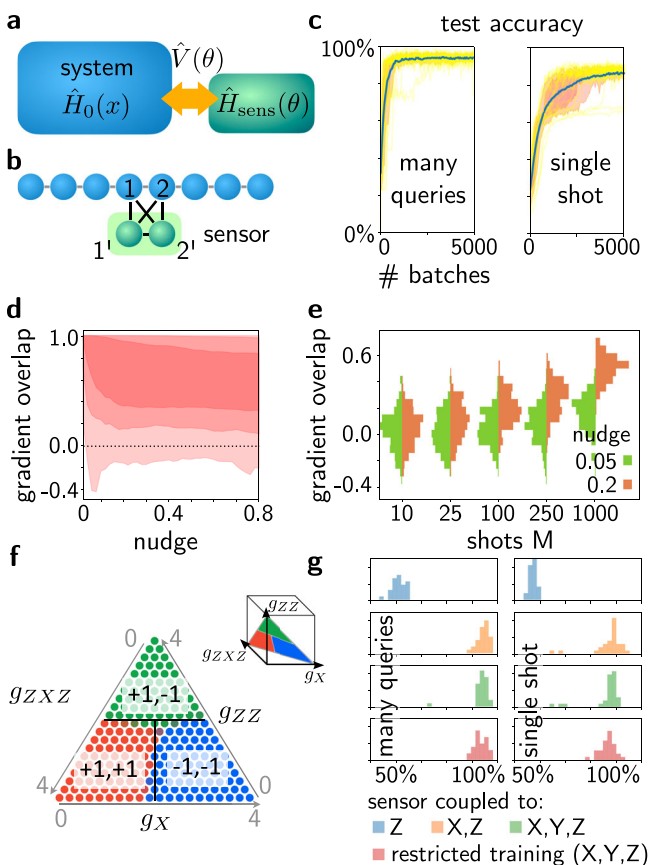

**Fig. 2 | Supervised learning: Learning recognition of quantum phases using QEP. a** Schematic of a trainable quantum sensor coupled to a system. **b** Specific example of a two-qubit sensor coupled to a 1D cluster transverse Ising Hamiltonian at two locations, where readout of $\hat{Z}_{1'}$ and $\hat{Z}_{2'}$ is supposed to indicate the phase. The 51 tuneable couplings are learned using QEP. **c** Evolution of test accuracy during supervised training on the whole phase diagram (for a chain of length $N = 8$). Multiple training runs (yellow), confidence intervals as areas (red; at 50% and 80%), and average accuracy (blue). Accuracies for 'many queries' (asking whether the maximum-probability detector outcome matches the correct result) and 'single shot' (probability to indicate the correct phase in single quantum measurement) [batches of 10 training samples; projection noise for $M = 10$ measurement shots per sample is accounted for; nudge parameter $\beta = 0.4$]. **d** Overlap of batch-averaged gradient estimate with true gradient direction, vs. nudge parameter. Positive overlap produces beneficial training updates. Confidence intervals (red, 95%, 80%, 50%) depict distribution over many batches (batch size 10, no measurement shot noise). **e** Gradient overlap histograms vs. measurement shots $M$, for two different nudge parameters (batch size 10). **f** Test of phase recognition: probabilities of measuring the trained sensor in one of the three different combinations of $Z_{1'}, Z_{2'}$ shown in orange/green/blue; true phase boundaries in black ($g_Z + g_{XX} + g_{ZXZ} = 4$). **g** Histogram of final test accuracies for repeated training runs (parameters as above), for a sensor that only couples to $\hat{Z}$ operators in the chain (or only to $\hat{X}$ and $\hat{Z}$), and for a sensor trained only on a small patch in the middle of each phase but tested throughout ("restricted").

moreover is far more general in the choice of systems—e.g. it does not require qubits as degrees of freedom, nor the ability to perform gates, nor any detailed knowledge of all aspects of the Hamiltonian.

Both for this task as well as others analyzed below, we choose the cluster Ising Hamiltonian (see e.g. ref. 53) as an illustrative quantum many-body system,

$$\hat{H}_0 = g_{ZXZ} \sum_j \hat{Z}_{j-1} \hat{X}_j \hat{Z}_{j+1} - g_{ZZ} \sum_j \hat{Z}_j \hat{Z}_{j+1} - g_X \sum_j \hat{X}_j, \quad (6)$$

in which $\hat{X}, \hat{Y}, \hat{Z}$ are the Pauli matrices. This model has three phases, including a topologically nontrivial one. We will now regard $x = (g_{ZXZ}, g_{ZZ}, g_X)$ as the input parameters.

To enable classification of the phases, we couple a sensor to the chain. This sensor is made of qubits that are coupled in all possible ways among each other (2-local, with terms like $\hat{X}_\alpha \hat{Y}_\beta$ or $Z_\alpha$ etc.) and to a limited region in the chain (couplings $\hat{Z}_\alpha \hat{X}_j$ etc., where $j$ is a spin in the chain). All of these couplings are tuneable. The sensor qubits are measured in the $Z$-basis, and the resulting configuration is supposed to announce the detected phase. Here we use a mean-squared-error loss function, although a categorical cross-entropy would be suitable as well. We find that even a small-scale sensor of only two qubits, coupled to two spins in the chain, has sufficient expressivity to properly learn the known phase diagram of the cluster Ising model (Fig. 2b, c), when trained in a supervised fashion using QEP. We also confirm that the quantum nature of the coupling is important. This can be ascertained by comparing to a sensor that is only allowed to couple to $\hat{Z}$ operators in the chain, which performs much more poorly (Fig. 2g). By contrast, allowing coupling to non-commuting obervables yields the observed good accuracy. We also investigate the influence of the coupling strength between system and sensor (SI). With the soft cutoff we enforced to keep the couplings small, the system learns to predict the correct phases even when it is only trained on patches within each phase. This ability to generalize is less pronounced without the cutoff on the couplings since, in this case, the system-sensor coupling generally become large during training which disturbs the system spins so that their state substantially deviates from the unperturbed state.

We investigate the influence of measurement shot noise ('Methods') which, in practice, will make the gradient estimates noisy with fluctuations $\sim 1/\sqrt{M}$ with $M$ the number of measurement shots. It is possible to counteract this effect by increasing the nudging strength, although this leads to a deviation from the linear response and hence the ideal gradients. As a result, there is a trade-off between shot noise and finite nudging which we analyze by inspecting the scalar product between the true and the estimated gradient (Fig. 2d, e).

Next, we demonstrate our method's generality by applying it to a more complex model. This is particularly relevant since QEP is a physics-based training technique which in actual future experimental implementations, would not be hampered by the exponential growth of the Hilbert space, unlike classical simulations. Specifically, we numerically simulate the training of a phase sensor coupled to a strongly correlated two-dimensional spin model, namely the honeycomb Kitaev model[54]

$$\hat{H}_{Kitaev} = \sum_\alpha \sum_{\mathbf{r}} J_\alpha \hat{\alpha}_{A, \mathbf{r}} \hat{\alpha}_{B, \mathbf{r} + \mathbf{r}_\alpha} \quad (7)$$

with $\alpha \in \{X, Y, Z\}$ and $\mathbf{r}_\alpha$ defined according to Fig. 3a. The model has three Abelian phases for $J_X \geq J_Y + J_Z$, $J_Y \geq J_Z + J_X$, or $J_Z \geq J_X + J_Y$, respectively, with other parameter values belonging to a non-Abelian phase. We train sensor spins coupled to $\hat{H}_{Kitaev}$ ('Methods') to distinguish the Abelian from the non-Abelian phases. Figure 3b shows the single-shot accuracy during the training. The best performing phase sensor achieved a single-shot accuracy of 89.4 % and a many-queries accuracy of 100 % on the evaluation set shown as inset of Fig. 3b ('Methods'). The corresponding phase diagram with the predicted phases is depicted in Fig. 3c.

## Unsupervised learning: phase exploration and sensitivity optimization

We now discuss two applications of unsupervised learning for which the gradients are used for optimization tasks.

**Phase exploration.** In this first example, we would like to explore the phase diagram of a quantum many-body system. In practice, this could

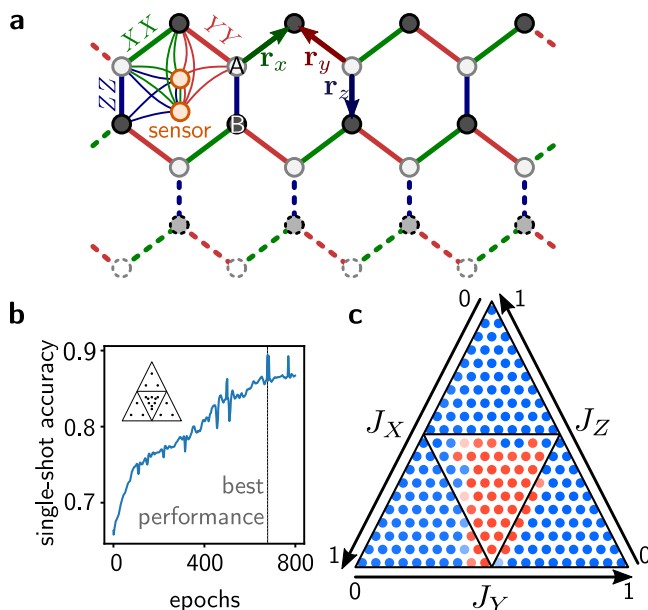

**Fig. 3 | Phase sensor for the honeycomb Kitaev model. a** In the Kitaev honeycomb model, spins are coupled with *XX*, *YY* and *ZZ* couplings on a hexagonal lattice. We couple a sensor (two spins) that will be trained to indicate the phase of the ground state. We simulate an 8 × 2 system with periodic boundary conditions (dashed lines; 'Methods'). **b** Single-shot accuracy on the evaluation set (inset) during the training. **c** Phase predicted by the best-performing sensor (blue: Abelian phase, red: non-Abelian). Following convention, $J_X + J_Y + J_Z = 1$ in this diagram.

be an interesting task for characterizing the capabilities of a quantum simulator when we would like to explore the phase diagram of a system which is partially unknown (e.g. some Hamiltonian terms are not known or cannot be tuned) and computationally hard to simulate classically. We can nevertheless ask whether phases exist that maximize (or minimize) certain expectation values and use QEP to explore the phase diagram and find such regimes. In this scenario, the relevant QEP-Hamiltonian of this unsupervised learning task is $\hat{H}(\theta, \nu)$, not containing any inputs $x$.

Figure 4 a illustrates the procedure: starting from an initial set of parameters $\theta$ (a point in the phase diagram), we optimize $\theta$ by computing the gradients with QEP to find the (potentially local) maximum of the expectation value of interest. The only difference in the QEP procedure here compared to the previous case is that the derivative of the simple loss function $\mathcal{L} = -y$ w.r.t. the output variable yields a trivial error signal of $-1$ which can then be inserted in the gradient calculation above according to Eq. (5).

Here, we first exemplify the procedure by examining a slice through the phase diagram of the cluster Ising Hamiltonian (6) when we fix $g_{ZXZ} = -0.5 \equiv$ const. We optimize the Néel order parameter $y \equiv \langle \hat{X}_0 \hat{X}_4 \rangle$, such that the loss function is simply $\mathcal{L}(y) \equiv -\langle \hat{X}_0 \hat{X}_4 \rangle$. We show two example trajectories in Fig. 4b. In both cases, the trajectories quickly converge to the $g_{ZZ} = 0$ line and then move along it, going into the paramagnetic phase. Accordingly, the loss function, Fig. 4c, rapidly decreases, with the decrease becoming slower as the trajectory moves along the $g_{ZZ} = 0$ line.

We also consider phase exploration for the honeycomb Kitaev model with an additional magnetic field in [1, 1, 1] direction breaking integrability

$$\hat{H} = \hat{H}_{\text{Kitaev}} + h \sum_{\beta} \sum_{\mathbf{r}} (\hat{X}_{\beta, \mathbf{r}} + \hat{Y}_{\beta, \mathbf{r}} + \hat{Z}_{\beta, \mathbf{r}}). \tag{8}$$

We use QEP to explore how the expectation value of the flux operator $y \equiv \langle \hat{W}_1 \rangle = \langle \hat{X}_1 \hat{Y}_2 \hat{Z}_3 \hat{X}_4 \hat{Y}_5 \hat{Z}_6 \rangle$, Fig. 4d, changes. While for $h = 0$, $\langle \hat{W}_1 \rangle = 1$

in the ground-state sector, this is no longer the case for $h \neq 0$ (see Methods). One can apply QEP to find the $J_\alpha$ that maximize $\langle \hat{W}_1 \rangle$ (Fig. 4e, f), and in an experiment this could be used for systems of any size, including arbitrary additional terms in the Hamiltonian.

We suggest that, in general, this approach could be an efficient technique for exploring higher dimensional phase diagrams which cannot simply be mapped out without considerable effort by sweeping all of the parameters.

**Sensitivity optimization.** In the second unsupervised learning application, the aim is to maximize the *derivative* of some expectation value of interest w.r.t. a certain parameter $\theta_j$. As Fig. 4g illustrates, in contrast to the previous examples, we now start with two points, $\theta^{(1)}$ and $\theta^{(2)}$, in the phase diagram at which we compute expectation values $y_{1,2} \equiv \langle \hat{A}_\ell \rangle |_{\theta^{(1,2)}}$ and maximize the slope calculated from the difference quotient of the output expectation values at these two points. Concretely, the corresponding loss function has the form

$$\mathcal{L}(y_1, y_2) = - \left| \frac{\langle \hat{A}_\ell \rangle |_{\theta^{(1)}} - \langle \hat{A}_\ell \rangle |_{\theta^{(2)}}}{\theta_j^{(1)} - \theta_j^{(2)}} \right|. \tag{9}$$

An appealing application of this procedure could be to find the optimal working point of sensors, such as magnetic field sensors.

To illustrate this scheme, we again consider the cluster Ising Hamiltonian (6) and optimize the slope of $\langle \hat{X}_0 \hat{X}_4 \rangle$ w.r.t. the parameter $g_X$, which is proportional to the magnetic field. Figure 4h shows two sets of example trajectories and the corresponding loss functions, Fig. 4i. We observe that the trajectories converge to the phase boundary, where the slope is largest. For the first run, Fig. 4j, we see that the trajectory moves along the phase boundary, since the slope is larger for smaller $g_{ZZ}$. We suggest that, in the future, this feature could be exploited to more generally map out phase boundaries.

## Discussion

We have exploited Onsager reciprocity to derive a quantum version of equilibrium propagation, which in its classical form is currently arguably the most widely studied general training technique for neuromorphic platforms. We have shown that this can be used successfully also for situations in which the input is effectively a quantum state (as in classifying quantum phases via supervised learning), as well as for unsupervised learning tasks (related to exploring the phase diagrams of quantum simulators or suited for optimizing sensing capabilities). In all of these cases, QEP can be applied even when the Hamiltonian is hard or impossible to simulate classically, since QEP extracts gradients via the physical response. In addition, it can be employed in cases in which only some aspects of the system are well characterized or accessible while other parts of the Hamiltonian are unknown. An important requirement is being able to reach the ground state experimentally, and we explained various options in the section on experimental requirements. Beyond the tasks analyzed in this manuscript, we can envisage further possibilities. One might train a tuneable quantum simulator to realize an arbitrary phase diagram, which is 'sketched', i.e., the phase diagram is prescribed in parts. Moreover, one might train it to approximately match the phase diagram and overall behaviour of another experimentally accessible quantum system, producing a quantum 'twin' and realizing the original promise of quantum simulations. A large variety of experimental platforms should be amenable to implementations of quantum equilibrium propagation, including highly tuneable systems based on trapped ions, Rydberg atoms, strongly interacting atoms in optical lattices, and superconducting qubit arrays. This will enable the transformation of many quantum simulators into learning machines, opening up a novel avenue for these intensively studied platforms.

During the final stage of completion of this manuscript and shortly before submission, two related preprints appeared on the

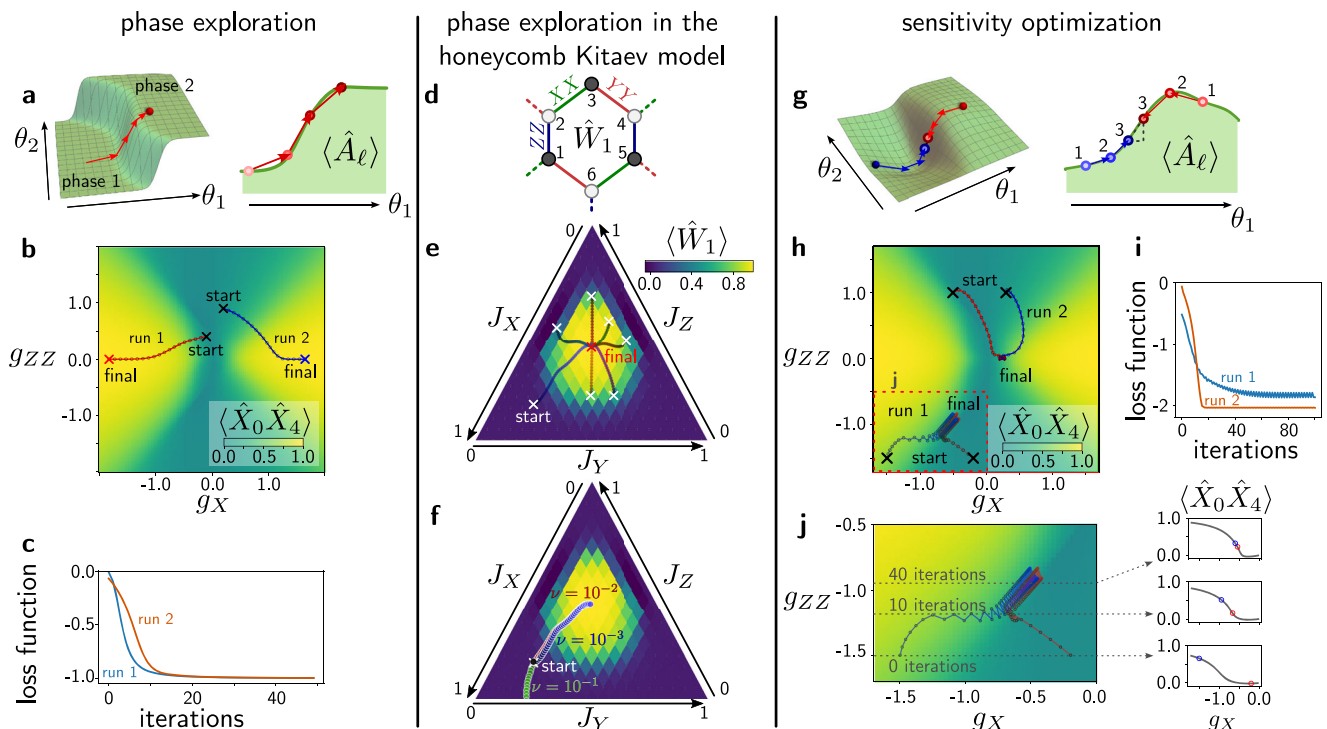

**Fig. 4 | Unsupervised learning applications: phase exploration and sensitivity optimization for sensing. a** Sketch of phase diagram exploration using QEP. An expectation value of interest, $\langle\hat{A}_\ell\rangle$, is maximised. **b** Two example trajectories in the phase diagram of the cluster Ising Hamiltonian (6), each starting at the position in the phase diagram marked with × and **c** the corresponding loss functions. The colours in **b** show $\langle\hat{X}_0\hat{X}_4\rangle$. **d-f** Phase exploration with QEP applied to the honeycomb Kitaev model in the presence of a magnetic field, Fig. 3a. **d** Flux operator $\hat{W}_1$ defined for the first plaquette. Without magnetic field, $\langle\hat{W}_1\rangle = 1$. We use QEP to find the parameters $J_\alpha$ for which $\langle\hat{W}_1\rangle$ is maximal. **e** Trajectories obtained with QEP and different initial conditions, converging to the same point ($\nu = 10^{-2}$). **f** Trajectories for different values of the nudging parameter $\nu$; larger values decrease the accuracy of the gradient, eventually preventing convergence. **g** Sketch of the concept of

sensitivity optimization. The derivative of a an observable of interest, $\langle\hat{A}_\ell\rangle$, w.r.t. a certain parameter (or w.r.t. a vector of parameters) is optimised. Concretely, this scheme can be used to devise optimal sensors or to find phase boundaries in a phase diagram. **h–j** Optimization of $\partial\langle\hat{X}_0\hat{X}_4\rangle/\partial g_X$ (magnetic field sensor). **h** Two example runs, each showing the trajectories of two points in the phase diagram of the cluster Ising Hamiltonian (6) and **i** the corresponding loss functions. The two points are initialized at the position marked with × and converge towards the phase boundary where the derivative w.r.t. $g_X$ is maximal. In the second run, **j** the trajectories first converge towards a phase boundary and then follow it, suggesting that the technique may be employed to trace out phase boundaries. The plots on the right show cuts through the phase diagram at various steps during the training.

arXiv[55,56], also introducing a quantum version of equilibrium propagation, with different use cases.

## Methods

### Supervised learning example

We considered a cluster Ising chain of $N = 8$ spins, and a sensor of 2 qubits, such that the total Hilbert space is 1024-dimensional. This allows efficient exact diagonalization using the Lanczos algorithm applied to sparse matrices, to find the ground state of the coupled system. The three considered output operators are projectors onto three states, each with a definite combination of the sensor operators $\hat{Z}_{1'}$ and $\hat{Z}_{2'}$, as shown in the figure; e.g., $\hat{P}_{1,-1} = (1 + \hat{Z}_{1'})(1 - \hat{Z}_{2'})/4$ projects onto the combination $(+1, -1)$ and would be used to indicate the ferromagnetic phase, which is reached when $g_{ZZ}$ dominates. The expectation value is correspondingly the probability to observe this particular combination in a projective measurement of these two operators. The training samples are drawn uniformly from the triangular phase diagram shown in the figure, in which (following the convention in the literature) we set the sum of all three coupling parameters to 4. The phase boundaries are known for this benchmark quantum many-body model[53], which allows us to provide the correct labels: in each phase, one of the projectors is 1, while the other two are 0 ('one-hot-encoding'), and the assignment of the three combinations of $(Z_{1'}, Z_{2'})$ to the three phases is defined in a fixed arbitrary way.

Gradients are obtained using the symmetric nudging procedure, by adding the nudged output operators $\sum_j \nu_\ell \hat{A}_\ell$ to the Hamiltonian, in

which in our case $\hat{A}_\ell = \hat{P}_\ell$ and $\ell \in \{(+1, +1), (+1, -1), (-1, -1)\}$. The ground state for the nudged Hamiltonian (for both signs of nudging) is recalculated using sparse Lanczos diagonalization. Interestingly, our experiments have shown that this is numerically more efficient than attempting to obtain the exact linear response using first-order perturbation theory, which involves solving a linear system of equations when applying $(E - \hat{H})^{-1}$ to the perturbed ground state. After the approximate gradient has been obtained using QEP, we use it inside an Adam adaptive gradient descent optimizer to update the parameters—which would be possible also for the QEP gradients obtained in real experiments. Just like for usual training of artificial neural networks, we group training samples into batches and actually employ the batch-averaged gradient. The learning rate employed in the numerical examples was set to 0.01.

Test accuracies are measured on a test set of 200 points that are also uniformly randomly distributed across the triangular space and which are fixed before the training run. We distinguish two measures of accuracy: In the 'many queries' version, we imagine that one would obtain the expectation values of the three projectors by simply measuring $\hat{Z}_{1'}$ and $\hat{Z}_{2'}$ simultaneously. After performing many measurement shots, this will yield the measurement probabilities for the three different combinations of outcomes $\{(+1, +1), (+1, -1), (-1, -1)\}$, corresponding to the three different projectors. We then assign as official outcome the phase whose associated projector has the largest expectation value (largest measurement probability). This is similar to how accuracy would be assessed for classical machine-learning

classification models, by identifying the label of maximum probability and comparing it against the true label. In the 'single shot' version, we imagine to run inference only once (equilibrating to the ground state once) and performing a single measurement of the two sensor operators. The outcome will be declared correct if the combination matches the correct label of the true underlying phase for this parameter combination. Single-shot performance is more difficult, but even so the training results show that single-shot accuracy can also reach relatively high values. Repetition, e.g. using three shots and taking a majority vote when possible, will quickly boost the accuracy (until it reaches the 'many queries' result in the limit of many shots).

**Influence of measurement shot noise.** An important general aspect of QEP training is the unavoidable projection shot noise encountered in any quantum experiment, in which expectation values are obtained by repeated measurements with individually discrete outcomes−in contrast to the classical situation.

For assessing the influence of shot noise, we replace the exact expectation values by numerical values that are drawn from a Gaussian distribution centred around that value, with the correct variance $\mathrm{Var}\hat{A}_\ell/M$, in which $M$ is the number of measurement shots (recall each of those will usually require a new equilibration, unless one adopts some of the strategies mentioned in the main text). Replacing the true distribution by a Gaussian is a reasonable approximation unless $M$ is very small.

Measurement shot noise hampers training, since the estimate of the gradient is noisy, with fluctuations $\sim 1/\sqrt{M}$, in which each of the $M$ shots requires a renewed equilibration. Fortunately, we find that this effect can be counteracted by employing an increased finite nudging strength $\beta$, effectively boosting the contrast in estimating the response of expectation values. Finite nudging, however, leads to a deviation from the linear response that would yield the ideal gradients. Therefore ultimately there is a sweet spot, balancing shot noise vs. nonlinearity of the response, to obtain optimized training convergence. This can be analyzed by inspecting the scalar product between the true gradient direction and the estimated gradient (Fig. 2d, e of the main text). In our numerical experiments, we find that the goal of minimizing the total number of experimental runs (i.e. number of training samples multiplied by number of shots per sample) is best achieved by keeping $M$ small and simply taking more samples. Apparently, this leads to more variety in the observed training data and better training performance. In experiments, the number of shots $M$ could also be reduced by coupling multiple sensors at different locations to the system, having them share their coupling values. Finally, in principle, there is an alternative to averaging over projective measurements, namely performing weak continuous measurements of the expectation values (a weakly coupled sensor is able to realize this). This could be performed without rethermalizing to the ground state.

**Aspects of scalability.** The numerical simulation results we show in the main text serve the purpose of illustrating the method QEP, but are naturally limited in the size of the systems we can treat, due to the exponential growth of the Hilbert space. This exponential growth of the numerical effort is also the reason why a directly physics-based training method such as QEP is attractive, since in the eventual application to experiments the method will precisely obviate the need for such challenging numerical simulations.

In the experiment, there are no constraints that would prevent the application of QEP to larger systems, as the number of measurements needed for QEP is independent of the system size. We note that in typical modern quantum simulator platforms care is taken to construct setups that require only a single experimental run to perform a measurement of all (commuting) observables simultaneously; a typical example being a single snapshot of ultracold atoms in an optical lattice revealing a projective measurement of the atom number in all lattice sites. Therefore, scaling up the system size does not in itself lead to an increase in the number of measurements. That being said, the total number of measurements is the product of the number of experiments that need to be done under varying conditions and the number of measurement shots per experiment. While straightforward parameter shift methods would require a number of experiments scaling linearly in the (potentially large) number of trainable parameters, the QEP method only requires a single linear-response experiment (or only a small fixed number of order 1) to physically extract the training gradient independent of system size.

Taking measurement shot-noise into account, one would expect that one has to perform around 100 measurement shots or more to get a reasonably accurate expectation. However, our numerical experiments reported in the main text show that the stochastic gradient descent still works very well even for a small number of shots. Regardless, the required number of shots does not increase with system size, and likewise the number of experiments remains the same (on order 1), independent of system size.

Finally, the required number of different training samples normally also does not increase with system size. In a supervised training scenario of the type envisaged in our manuscript, e.g., for recognizing the phase of the quantum many-body ground state, the number of training samples will determine how densely one can sample the phase diagram of the system. However, larger system sizes do not require a larger number of training samples, since the learning tasks remains essentially the same (distinguishing a finite number of potential phases).

## Technical details of the phase sensor for the Kitaev honeycomb model

We simulate a lattice of $8 \times 2$ spins which is coupled to two additional sensor spins, Fig. 3a, i.e., 8 plaquettes, under periodic boundary conditions. Since we perform exact diagonalisation to compute expectation values in the ground state, we choose a system size that allows to perform the calculations within a reasonable time frame. We note that the envisioned implementation of the system with a quantum simulator would not have the same limitations. We choose this particular $8 \times 2$ lattice to keep the impact of finite size effects as small as possible. In particular, without the correct boundary conditions, contrary to what is expected from the exact solution in the thermodynamic limit $\langle \hat{W}_1 \rangle \neq 1$ at finite system size. This is because the quantization of $\langle \hat{W}_1 \rangle$ is based on Lieb's theorem which only applies in the limit of an infinite system or with the correct, twisted periodic boundary conditions at finite size[57]. Limiting the total number of system spins to 16, we chose to simulate a system of $8 \times 2$ spins for which the normal periodic boundary conditions correspond to the twisted boundary conditions needed so that Lieb's theorem holds and $\langle \hat{W}_1 \rangle$ is quantised at finite system size.

We consider couplings between the two sensor spins (all possible combinations of Paulis) as well as couplings to the system spins of the Kitaev model via the operators $\hat{X}_{A,\mathbf{r}}\hat{X}_{B,\mathbf{r}+\mathbf{r}_x}$, $\hat{Y}_{A,\mathbf{r}}\hat{Y}_{B,\mathbf{r}+\mathbf{r}_y}$, $\hat{Z}_{A,\mathbf{r}}\hat{Z}_{B,\mathbf{r}+\mathbf{r}_z}$ of one plaquette, Fig. 3a. See the Supplementary Material for further details about the numerical implementation.

## Different approaches for nudging

In classical EP, the idea is to add the loss function $\mathcal{L}(y, y^{\mathrm{target}}(x))$ to the energy, multiplied by $\beta$, in which $\beta$ is a small constant. Instead, in the main text we advocated simply adding $\sum_\ell \nu_\ell \hat{A}_\ell$ to the Hamiltonian, in which $\hat{A}_\ell$ are the output operators and $\nu_\ell = \beta \partial \mathcal{L}/\partial y_\ell$ is the nudge force, since for small $\beta$ this produces the force needed to elicit the linear response required for the gradient. If we were instead to translate directly the classical EP prescription to the quantum Hamiltonian, we could add to the Hamiltonian a term $\beta \mathcal{L}(\hat{y}, y^{\mathrm{target}}(x))$, with the operator version of the outputs, $\hat{y}_\ell = \hat{A}_\ell$ for $\ell \in \mathcal{S}_{\mathrm{out}}$, replacing the expectation values $y_\ell$. Expanding this term to linear order in $\hat{A}_\ell$,

we would obtain the same result as our ansatz. The higher-order terms in $\beta\mathcal{L}$ would lead to further corrections to the Hamiltonian, and depending on the quantum fluctuations in $\hat{A}_\ell$ these might be as large as the linear-order term itself (e.g. for mean-squared-error loss functions, these could correspond to a stiffening of the potential acting on $\hat{A}_\ell$). It is difficult to assess a priori the effect of these fluctuations arising from such an ansatz $\beta\mathcal{L}(\hat{y}, y^{\text{target}}(x))$. Furthermore, these higher-order terms might also be more challenging to implement experimentally, depending on the shape of $\hat{A}_\ell$. Since they are not needed to evaluate the gradient, we chose the procedure explained in the main text, adding only linear terms to the Hamiltonian. It would be interesting in the future to compare the various approaches for finite nudging, when $\beta$ is not small (which is actually the case for the numerical experiments).

### Unsupervised learning examples

**Phase exploration.** We again consider a cluster Ising Hamiltonian of 10 spins, and search for the points in the phase diagram that maximize $\langle\hat{X}_0\hat{X}_4\rangle$, hence, $\hat{X}_0\hat{X}_4$ is the output operator. This is expected to be maximized in the paramagnetic phase. Initial parameters will typically be chosen randomly. For illustrative purposes, we manually select the initial parameters, i.e. the starting points of the trajectories in Fig. 4b. Concretely, the two starting points are $g_X = -0.1, g_{ZZ} = 0.4$ (run 1) and $g_X = 0.9, g_{ZZ} = 0.9$ (run 2). We obtain gradients according to the QEP procedure by switching on the coupling to the output operator $\nu\hat{X}_0\hat{X}_4$. As in the supervised learning example, the ground state is computed using sparse Lanczos diagonalization. For the results shown in Fig. 4b, both the learning rate and the nudge parameter $\nu$ were set to 0.1.

**Sensitivity optimization.** In the second unsupervised learning example, we search for the largest slope of $\langle\hat{X}_0\hat{X}_4\rangle$ w.r.t. $g_X$ in the phase diagram of the cluster Ising Hamiltonian (6) with 10 spins by optimizing the loss function (9). Concretely, we consider two different values of $g_X^{(1,2)}$ while the other parameters are the same ($g_{ZZ}$ is trained while $g_{ZXZ} = -0.5 \equiv$ const.).

During one step of the training, we update $g_X^{(1)}, g_X^{(2)}$ and $g_{ZZ}$. To obtain the necessary gradients, we need to compute the following derivatives (for brevity we denote $\partial/\partial\theta_j$ by $\partial_{\theta_j}$)

$$\partial_{g_X^{(1)}}\mathcal{L} = \varepsilon\,\frac{\partial_{g_X^{(1)}}\langle\hat{X}_0\hat{X}_4\rangle|_{g_X^{(1)}}}{|g_X^{(1)} - g_X^{(2)}|} + \text{sgn}\,(g_X^{(1)} - g_X^{(2)})\frac{\mathcal{L}}{|g_X^{(1)} - g_X^{(2)}|} \tag{10}$$

$$\partial_{g_X^{(2)}}\mathcal{L} = -\varepsilon\,\frac{\partial_{g_X^{(2)}}\langle\hat{X}_0\hat{X}_4\rangle|_{g_X^{(2)}}}{|g_X^{(1)} - g_X^{(2)}|} - \text{sgn}\,(g_X^{(1)} - g_X^{(2)})\frac{\mathcal{L}}{|g_X^{(1)} - g_X^{(2)}|} \tag{11}$$

$$\partial_{g_{ZZ}}\mathcal{L} = \varepsilon\,\partial_{g_{ZZ}}\frac{\left(\langle\hat{X}_0\hat{X}_4\rangle|_{g_X^{(1)}} - \langle\hat{X}_0\hat{X}_4\rangle|_{g_X^{(2)}}\right)}{|g_X^{(1)} - g_X^{(2)}|} \tag{12}$$

with $\varepsilon \equiv \text{sgn}\,[\langle\hat{X}_0\hat{X}_4\rangle|_{g_X^{(1)}} - \langle\hat{X}_0\hat{X}_4\rangle|_{g_X^{(2)}}]$. In all of the expressions, we use QEP to extract $\partial_{\theta_j}\langle\hat{X}_0\hat{X}_4\rangle|_{g_X^{(1,2)}}$ with $\theta_j \in \{g_X^{(1)}, g_X^{(2)}, g_{ZZ}\}$.

To that end, as outlined in the main text, we approximate the derivative by comparing the nudged and the free expectation value

$$\frac{\partial}{\partial\theta_j}\langle\hat{X}_0\hat{X}_4\rangle \approx \frac{\langle\hat{X}_0\hat{X}_4\rangle|_{\nu=\beta\varepsilon} - \langle\hat{X}_0\hat{X}_4\rangle|_{\nu=0}}{\beta}. \tag{13}$$

As before, to evaluate the expectation value in the nudged phase, we couple to the output operator by adding $\nu\hat{X}_0\hat{X}_4$ to the Hamiltonian. Ground states in the free and the nudged phase are again computed using sparse Lanczos diagonalization.

To illustrate the procedure, we consider two different starting points in the phase diagram: $g_X^{(1)} = -0.2, g_X^{(2)} = -1.5, g_{ZZ} = -1.5$ (run 1; each point starts in a different phase) and $g_X^{(1)} = -0.5, g_X^{(2)} = 0.3, g_{ZZ} = 1.0$ (run 2; both points are in the same phase). Both the learning rate and the nudging parameter were set to 0.1.

**Further details about phase exploration with the Kitaev honeycomb model.** In the limit $h = 0$, the Kitaev honeycomb model, Eq. (8) can be mapped to a bi-linear Majorana fermion model and is exactly solvable[54]. In that case, the flux operator defined on one plaquette, e.g. $\hat{W}_1 = \hat{X}_1\hat{Y}_2\hat{Z}_3\hat{X}_4\hat{Y}_5\hat{Z}_6$, has eigenvalues $\pm1$ with the $+1$ eigenvalue corresponding to the ground state sector in which $\langle\hat{W}_1\rangle = 1$. Switching on the magnetic field $h \neq 0$, the model is not solvable and $\langle\hat{W}_1\rangle$ is no longer quantized, so we use QEP in the main text to explore how $\langle\hat{W}_1\rangle$ changes as a function of $J_\alpha$; in particular, we fix $h = 0.05 \equiv$ const. and are looking for the parameters $J_\alpha$ that maximize $y \equiv \langle\hat{W}_1\rangle$. Note that any other plaquette would yield the same behaviour due to translational symmetry.

## Data availability

The numerical data produced in this work are available from the repository at https://github.com/ClaraWanjura/QuantumEP.

## Code availability

The code to generate the results discussed in this work is available from the GitHub repository[58] at https://github.com/ClaraWanjura/QuantumEP and Zenodo[59] at https://doi.org/10.5281/zenodo.15741281.

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

## Acknowledgements

We acknowledge funding via the German Research Foundation (DFG, Project-ID 429529648-TRR 306 QuCoLiMA, 'Quantum Cooperativity of Light and Matter'). This research is part of the Munich Quantum Valley, which is supported by the Bavarian state government with funds from the Hightech Agenda Bayern Plus.

## Author contributions

Both authors, C.C.W. and F.M., developed the idea, ran the numerical experiments and wrote the manuscript.

## Funding

## Competing interests

The authors declare no competing interests.
