## [Transparent Peer Review file · Nature Communications]

Quantum Equilibrium Propagation for efficient training of quantum systems based on Onsager reciprocity

Corresponding Author: Dr Clara Wanjura

Version 0:

Reviewer comments:

Reviewer #1

(Remarks to the Author)

Please see the pdf file for better viewing of our comments.

In this paper, the authors propose a method for training neuromorphic quantum systems. The method, called quantum equilibrium propagation (QEP), is a quantum version of the widely studied equilibrium propagation (EP) technique that exploits Onsager reciprocity to apply the training procedure to quantum Hamiltonians. The feasibility and applicability of the QEP approach are demonstrated via numerical simulations by solving small-scale supervised and unsupervised learning tasks in many-body quantum physics and sensing.

In our opinion, the current manuscript at the present stage does not have enough merit to be accepted in a prestigious journal such as Nature Communications.

To be fair to the authors, we provide detailed questions, comments, and doubts that should be cleared for future submission. Our questions and comments are listed in the order of appearance in the manuscript.

\begin{enumerate}

\item Under the introduction and the Discussion, the authors mentioned multiple times about the recent popularity of neuromorphic computing.

But, those reasoning does not go very well with what the authors presented in the main text.

In that sense, the manuscript is not clearly written.

What is the main selling point here?

Do the authors like to aid in solving some or a part of the \textbf{classical} neuromorphic computing or the \textbf{quantum} neuromorphic computing?

If the focus was in ``quantum neuromorphic one'', how the proposed method is helpful in that system architecture?

Is it part of the quantum memristor circuit?

\item The numerical demonstrations are performed for very small systems. Do the authors have any comments on the scalability of the method for large systems? For example, regarding the number of measurements needed or the influence of training samples.

\item In addition to the point 2 above, we do not think that the cost estimate of the training parameters based on the Onsager reciprocity is of the order $\$1\$$ as suggested by the authors under the figure 1 caption as well as Section II Results.

To be specific, let us look at Eq.(3), the Onsager reciprocity.

It simply means that the susceptibility matrix χ_{ij} is symmetric.

To get the gradient for all the parameters, as we understand based on Eq.(3), instead of finding gradients for n^2 times, we would require $n(n+1)/2$, assuming the square matrix with the size $n \times n$.

Hence, the claim of order $\$1\$$ is not clear to us and we believe it is not correct.

\item The last paragraph of Sec IIA says ``Onsager reciprocity teaches us how to translate the classical equilibrium propagation approach to quantum devices."

It implies that one definitely needs ``Onsager reciprocity" to realize quantum equilibrium propagation. Please clarify.

\item Under the Sec IIB, right hand column of page 3, the 'nudged phase' requires an or many additional Hamiltonian terms

composed of $\sum_i \nu_i A_i$. Also in the example case on page 5, the authors mentioned that "Inspecting the solution also reveals a surprise: the sensor-system couplings are not weak..."

These two facts bother us a lot.

This is beyond our physics understandings.

As the main idea here is to use the ground state equilibrium propagation, such strong sensor-system coupling/s would definitely change the original quantum system ground state.

What is/are main condition/s to make it work?

One of the weaknesses of the proposed method is the requirement for the quantum system to reach the ground state.

As explained in the manuscript, the efficient preparation of the ground state of arbitrary complex quantum many-body Hamiltonians is generally a very difficult computational task. Although the authors certainly discuss this issue in section II.C, it would be beneficial to include some comments in the discussion in section III when stating the applicability of QEP: "In all these cases, QEP can be applied even [...] while other parts of the Hamiltonian are unknown."

The manuscript would benefit from a comparison with the current training methods available for neuromorphic quantum systems.

Without such comparisons, we can't really see how efficient the proposed method is.

To conclude, there are many technical uncertainties in the present manuscript that have to be fixed.

The main motivation, introduction and discussion are not clear. In fact, they are very ad-hoc and disjointed with the main results.

Hence, we can't approve its publication at Nature Communications.

(Remarks on code availability)

NIL

Reviewer #2

(Remarks to the Author)

(Remarks on code availability)

Reviewer #3

(Remarks to the Author)

In this manuscript, the authors demonstrate a direct connection between equilibrium propagation — training method for neural networks that focuses on achieving and maintaining equilibrium states (rather than relying solely on gradient-based optimization) — and Onsager reciprocity which asserts a relationship between different thermodynamic forces and fluxes in a system close to equilibrium.

The authors then use this revealed — and I will also add interesting — connection to derive a quantum version of equilibrium propagation which they then test of specific one-dimensional quantum Ising model in the context of both supervised and unsupervised learning and show that their method may be successfully used for phase detection, which is one the main proposed uses of neural network in the context of quantum many body systems.

The paper is skillfully written and was made very clear to understand. The derivation, results and methods are explained clearly and as far as I could tell are sound.

The method, that was indeed demonstrated to work, if it does indeed work will undoubtedly become a useful tool at the interface of quantum systems and machine learning. I have but one "major" quibble which is — and this is true for all studies presenting methods of this nature — results are always almost exclusively presented on one-dimensional systems which are known to be much easier to solve than two- and more-dimensional systems. As the authors I am sure are well aware, it is the high-dimensional quantum systems that are the real goal of phase detection and yet it is unclear whether the presented method is expected to perform well on scientifically interesting models. If so, then no doubt that the proposed method deserves to be published in a high impact journal such as Nature Communications. The reader will surely appreciate a discussion or even seeing results pertaining to the performance of the proposed method on more challenging quantum many-body models.

(Remarks on code availability)

The code is written skillfully and contains sufficient documentation.

Version 1:

Reviewer comments:

Reviewer #1

(Remarks to the Author)

In light of the recent referees' correspondences by the authors, with additional data on learning quantum phases of Kitaev model, I find that the manuscript is on par the quality of nature comm. Thus, I will recommend its publication.

Congratulations!

(Remarks on code availability)

Reviewer #2

(Remarks to the Author)

(Remarks on code availability)

Reviewer #3

(Remarks to the Author)

The authors have devoted muchtime and effort to address all referees remarks and concerns in full. They also addressed my concern that the method was shown to work only on a 1D system by considering a 2D system. There, unfortunately, the results were quite limited as they are only simulations of a quantum algorithm.

The proposed algorithm is still very interesting. The results are promising yet I would say not definitive.

(Remarks on code availability)

N/A

Reply for ‘Quantum Equilibrium Propagation for efficient training of quantum systems based on Onsager reciprocity’

List of Changes

In response to the referee reports, we made the following changes to our manuscript (for more details please see the detailed reply and the manuscript):

- Following reviewer 3’s demand to investigate more complex systems, we performed extensive new simulations on an entirely different model—a two-dimensional model, the Kitaev honeycomb model. We both trained a phase sensor to distinguish Abelian vs. non-Abelian phases as well as performed phase exploration with a non-integrable modification of the model.
- Based on a suggestion of reviewers 1/2, we performed extensive new simulations for the Ising model sensor, limiting the strength of the system-sensor coupling. This has also improved the generalizability of our sensor. We replaced the data in Figure 2 **c**, **f** and **g** with our new results and added a section providing further details to the Supplementary Information in which we discuss the influence of this system-sensor coupling strength as well as the cutoff we employ during training to limit this strength.
- To better clarify the context of our work, we added panel **c** to Figure 1 in which we provide an overview of different classical and quantum approaches to neuromorphic computing.
- Addressing a comment of reviewers 1/2, we extended our discussion on scalability and shot noise, providing more details in a new Methods section.
- In response to reviewers 1/2, we added a derivation of Onsager reciprocity to the Supplementary Information.
- Elaborating on a comment of reviewers 1/2, we added a section on a comparison to other quantum machine learning approaches to the Supplementary Information.
- Answering a question by reviewers 1/2, we clarified in the main text why our method to extract all gradients scales so favorably with the number of parameters.
- Following a suggestion of reviewers 1/2, in the Discussion section, we added a reference to our experimental requirements section which contains a detailed discussion of schemes to attain the equilibrium state.

In the following, we reply to the reviewer remarks on a point-by-point basis. The report by the reviewer is typeset in **blue**, our reply in black and changes to the manuscript in **green**.

Reviewer #1 (Remarks to the Author):

In this paper, the authors propose a method for training neuromorphic quantum systems. The method, called quantum equilibrium propagation (QEP), is a quantum version of the widely studied equilibrium propagation (EP) technique that exploits Onsager reciprocity to apply the training procedure to quantum Hamiltonians. The feasibility and applicability of the QEP approach are demonstrated via numerical simulations by solving small-scale supervised and unsupervised learning tasks in many-body quantum physics and sensing.

In our opinion, the current manuscript at the present stage does not have enough merit to be accepted in a prestigious journal such as Nature Communications. To be fair to the authors, we provide detailed questions, comments, and doubts that should be cleared for future submission. Our questions and comments are listed in the order of appearance in the manuscript.

1. Under the introduction and the Discussion, the authors mentioned multiple times about the recent popularity of neuromorphic computing. But, those reasoning does not go very well with what the authors presented in the main text. In that sense, the manuscript is not clearly written. What is the main selling point here? Do the authors like to aid in solving some or a part of the **classical** neuromorphic computing or the **quantum** neuromorphic computing? If the focus was in "quantum neuromorphic one", how the proposed method is helpful in that system architecture? Is it part of the quantum memristor circuit?

We thank the reviewer for the question.

Since our method is inspired by a general training method originally developed for classical neuromorphic computing, we decided to start the manuscript by reviewing some of the literature surrounding this topic. Specifically, equilibrium propagation was first developed to train energy-based classical neuromorphic systems. By realising that, in fact, Onsager reciprocity lies at the heart of Equilibrium Propagation (this is already a new insight), we are then able to generalise the training method to the quantum-mechanical realm.

When applied to the quantum realm, the ability to experimentally extract gradients efficiently suddenly opens up completely new possibilities and applications which of course differ from the applications typically discussed in the context of classical neuromorphic computing. New applications include the examples we discuss in the main text, i.e., applying quantum equilibrium propagation to quantum simulation experiments for the purpose of ground state classification, phase exploration, phase boundary detection and sensitivity optimization. Quantum equilibrium propagation can therefore help us gain more fundamental insight into many-body systems that are numerically hard to simulate.

Quantum Equilibrium Propagation represents an entirely new approach to perform training and optimization in experiments on complex quantum systems. It is very different from the only other generic 'quantum machine learning' technique which relies on variational quantum circuits, where one presupposes the availability of a quantum computer (a 'digital approach'), which is a much more challenging requirement than what will be needed for QEP.

We imagine that part of the confusion may also have arisen from the fact that e.g. the word 'neuromorphic computing' has been used in different ways throughout the literature.

Action taken: To avoid misunderstandings and to clarify how we view the entire field of neuromorphic computing, we added a schematic new figure (Fig. 1 c), which displays an overview of different areas classified along two aspects (classical vs. quantum, analog vs digital) and highlights the unique spot that QEP will occupy. For convenience, we provide the updated figure and text below:

Figure 1. The concept of quantum equilibrium propagation.

The goal is to efficiently and in a physical way obtain the gradient of some loss function (depending on expectation values measured at the “output” degrees of freedom of a quantum system) with respect to tuneable parameters. **a** Rather than shifting N parameters separately and measuring the output expectation value for each shift (parameter-shift method), Onsager reciprocity dictates that the same information can be extracted by **b** shifting, i.e. nudging, only the parameters coupling to the output observables and (in a single go) measuring the response of all N operators coupled to the training parameters. This procedure, termed quantum equilibrium propagation, is more efficient as it requires only a single response experiment (or at most a small number of order 1, when some non-commuting observables have to be measured), whereas the parameter shift method requires a number of experiments scaling linearly with the number of parameters. **c** Overview inspired by [16] of digital and analogue neuromorphic computing schemes and platforms in the classical and quantum regime. Quantum equilibrium propagation can be applied to analogue quantum platforms such as quantum simulators with atoms and superconductors.

Furthermore, we added the following sentence at the end of the introduction:

... The major research efforts in this domain are, however, spent on variational quantum circuits,

which require fully controllable digital quantum computing platforms for implementation (possibly even fault-tolerant), imposing resource demands that go significantly beyond what we are going to assume here. We provide an overview of digital and analogue neuromorphic computing approaches in the classical and quantum domain in Fig. 1 c.

2. The numerical demonstrations are performed for very small systems. Do the authors have any comments on the scalability of the method for large systems? For example, regarding the number of measurements needed or the influence of training samples.

We thank the referee for this question. First, we recall the important point that QEP is proposed as a physics-based training method to be applied to experimental systems. Only for the purpose of illustrating the method in the present theory manuscript we have to resort to numerical simulations. Such simulations are naturally limited in the size of the systems we can treat, due to the exponential explosion of the Hilbert space. This exponential explosion of the numerical effort is of course also the reason why a directly physics-based training method is attractive, since in the eventual application to experiments the method will precisely obviate the need for such challenging simulations.

Regarding the number of measurements needed, this is another strong point of the method. In the experiment, there are no constraints that would prevent the application of QEP to larger systems, as the number of measurements needed for QEP is independent of system size. We note that in typical modern quantum simulator platforms care is taken to construct setups that require only a single experimental run to perform a measurement of all (commuting) observables simultaneously; a typical example being a single snapshot of ultracold atoms in an optical lattice revealing a projective measurement of the atom number in all lattice sites. Therefore, scaling up the system size does not in itself lead to an increase in the number of measurements. That being said, the total number of measurements is the product of the number of experiments that need to be done under varying conditions and the number of measurement shots per experiment. While straightforward parameter shift methods would require a number of experiments scaling linearly in the (potentially large) number of trainable parameters, the QEP method proposed in our manuscript only requires a single linear-response experiment (or only a small fixed number of order 1) to physically extract the training gradient. We explain this point again also in the answer to the next question. Regarding the number of shots per experiment, one would usually expect that this number has to be on the order of 100 or more to get reasonably accurate expectation values despite measurement shot noise, but our numerical experiments show that the stochastic gradient descent still works very well even for a small number of shots. In any case, the required number of shots does not increase with system size, and likewise the number of experiments remains the same (of order 1), regardless of system size.

Regarding the number of different required training samples, we can also state that this does not normally increase with system size. In a supervised training scenario of the type envisaged in our manuscript, e.g., for recognizing the phase of the quantum many-body ground state, the number of training samples will determine how densely one can sample the phase diagram of the system. However, larger system sizes do not require a larger number of training samples, since the learning tasks remains essentially the same (distinguishing a finite number of potential phases).

Action taken:

- (a) We added a new section to Methods in which we discuss the scalability of QEP.
- (b) We moved the discussion of measurement shot noise from the main text into the Methods (to adhere to Nature Communications' length restrictions) and expanded on the required number of measurements.

For convenience, we report the new/ moved subsections we added to Methods below.

Influence of measurement shot noise

An important general aspect of QEP training is the unavoidable projection shot noise encountered in any quantum experiment, in which expectation values are obtained by repeated measurements with individually discrete outcomes—in contrast to the classical situation.

For assessing the influence of shot noise, we replace the exact expectation values by numerical values that are drawn from a Gaussian distribution centered around that value, with the correct variance $\text{Var}\hat{A}_\ell/M$, in which M is the number of measurement shots (recall each of those will usually require a new equilibration, unless one adopts some of the strategies mentioned in the main text). Replacing the true distribution by a Gaussian is a reasonable approximation unless M is very small.

Measurement shot noise hampers training, since the estimate of the gradient is noisy, with fluctuations $\sim 1/\sqrt{M}$, in which each of the M shots requires a renewed equilibration. Fortunately, we find that this effect can be counteracted by employing an increased finite nudging strength β , effectively boosting the contrast in estimating the response of expectation values. Finite nudging, however, leads to a deviation from the linear response that would yield the ideal gradients. Therefore ultimately there is a sweet spot, balancing shot noise vs. nonlinearity of the response, to obtain optimized training convergence. This can be analyzed by inspecting the scalar product between the true gradient direction and the estimated gradient (Fig. 2 **d,e** of the main text). In our numerical experiments, we find that the goal of minimizing the total number of experimental runs (i.e. number of training samples multiplied by number of shots per sample) is best achieved by keeping M small and simply taking more samples. Apparently, this leads to more variety in the observed training data and better training performance. In experiments, the number of shots M could also be reduced by coupling multiple sensors at different locations to the system, having them share their coupling values. Finally, in principle, there is an alternative to averaging over projective measurements, namely performing weak continuous measurements of the expectation values (a weakly coupled sensor is able to realize this). This could be performed without rethermalizing to the ground state.

Aspects of scalability

The numerical simulation results we show in the main text serve the purpose of illustrating the method QEP, but are naturally limited in the size of the systems we can treat, due to the exponential growth of the Hilbert space. This exponential growth of the numerical effort is also the reason why a directly physics-based training method such as QEP is attractive, since in the eventual application to experiments the method will precisely obviate the need for such challenging numerical simulations.

In the experiment, there are no constraints that would prevent the application of QEP to larger systems, as the number of measurements needed for QEP is independent of the system size. We note that in typical modern quantum simulator platforms care is taken to construct setups that require only a single experimental run to perform a measurement of all (commuting) observables simultaneously; a typical example being a single snapshot of ultracold atoms in an optical lattice revealing a projective measurement of the atom number in all lattice sites. Therefore, scaling up the system size does not in itself lead to an increase in the number of measurements. Overall, the total number of measurements is the product of the number of experiments that need to be done under varying conditions and the number of measurement shots per experiment. While straightforward parameter shift methods would require a number of experiments scaling linearly in the (potentially large) number of trainable parameters, the QEP method only requires a single linear-response experiment (or only a small fixed number of order 1) to physically extract the training gradient independent of system size.

Taking measurement shot-noise into account, one would expect that one has to perform around 100 measurement shots or more to get a reasonably accurate expectation. However, our numerical experiments reported in the main text show that the stochastic gradient descent still works very well even for a small number of shots. Regardless, the required number of shots does not increase with system size, and likewise the number of experiments remains the same (of order 1), independent of system size.

Finally, the required number of different training samples normally also does not increase with system size. In a supervised training scenario of the type envisaged in our manuscript, e.g., for recognizing the phase of the quantum many-body ground state, the number of training samples will determine how densely one can sample the phase diagram of the system. However, larger system sizes do not require a larger number of training samples, since the learning tasks remains essentially the same (distinguishing a finite number of potential phases).

Instead of the paragraph on shot noise in the main text, we now provide a shortened paragraph, referring to the Methods for further details. The new main-text paragraph is reproduced below.

We investigate the influence of measurement shot noise (Methods) which, in practice, will make the gradient estimates noisy with fluctuations $\sim 1/\sqrt{M}$ with M the number of measurement shots. It is possible to counteract this effect by increasing the nudging strength, although this leads to a deviation from the linear response and hence the ideal gradients. As a result, there is a trade-off between shot noise and finite nudging which we analyze by inspecting the scalar product between the true and the estimated gradient (Fig. 2 **d,e**).

In the revised manuscript, we now also emphasize in the main text the point that the number of measurements required for QEP is independent of system size. Specifically in the caption of Fig. 1:

This procedure, termed quantum equilibrium propagation, is more efficient as it requires only a single response experiment (or at most a small number of order 1, when some non-commuting observables have to be measured) independent of the system size, whereas the parameter shift method requires a number of experiments scaling linearly with the number of parameters.

3. In addition to the point 2 above, we do not think that the cost estimate of the training

parameters based on the Onsager reciprocity is of the order 1 as suggested by the authors under the figure 1 caption as well as Section II Results. To be specific, let us look at Eq.(3), the Onsager reciprocity. It simply means that the susceptibility matrix χ_{lj} is symmetric. To get the gradient for all the parameters, as we understand based on Eq.(3), instead of finding gradients for n^2 times, we would require $n(n+1)/2$, assuming the square matrix with the size $n \times n$. Hence, the claim of order 1 is not clear to us and we believe it is not correct.

We thank the referee for this important question. Indeed, Onsager reciprocity implies that the susceptibility is symmetric. However, the key is that with our scheme, we can exploit this symmetry in order to extract all components of the gradient vector in parallel, giving rise to the order 1 effort, as we explain in the following.

Let us for simplicity first assume that there is only one output operator, for example in a binary classification task. With the straightforward parameter-shift method, Fig. 1 a, one has to shift one trainable parameter at a time, i.e. perform a new experiment for each of the N parameters. In each of these experiments, one measures the resulting change in the output operator expectation value. The entirety of these N experiments then provides the gradient needed for training. Rephrasing this in the language of the susceptibility matrix, if our output operator corresponds to some fixed single value of the index $\ell = \ell^*$ in the susceptibility matrix $\chi_{\ell,j}$, then we need to do an individual linear-response experiment for each of the N possible values of j , to obtain all entries $\chi_{\ell^*,j}$ which together will give us the training gradient. Therefore, the total time needed to estimate a gradient based on this method would scale with the number N of parameters. In contrast, making use of Onsager reciprocity we have $\chi_{\ell,j} = \chi_{j,\ell}$. This reveals that obtaining all the values of $\chi_{\ell^*,j}$ is equivalent to obtaining all the values of χ_{j,ℓ^*} , for arbitrary j . Importantly, the latter can be obtained in a *single* linear-response experiment. In this experiment, we only have to shift (nudge) the output operator at $\ell = \ell^*$ once and then measure each of the resulting shifts in the expectation values of the N operators labeled by j . The key is that this can be done *in parallel*, i.e. at the same time, if the operators commute, by performing measurements all across the quantum many-body system after nudging, thus indeed requiring only a single experiment. If the operators do not commute, additional experiments are necessary, for example if both the values of x- and y-components of a spin have to be measured in a spin system at every lattice site. However, this only amounts to a small finite fixed factor (two in this example), and the total number of experiments will therefore still be of order 1. The overall reduction in the number of experiments needed, thanks to exploiting Onsager reciprocity, is therefore a substantial factor, of order N .

More generally, if the number of output operators is N_{out} , the number of experiments would at first sight scale with N_{out} , repeating the procedure above for various values of ℓ^* . However, this again can be reduced to order 1, if we apply nudge forces simultaneously to all output operators, with the force amplitudes determined by the so-called error vector $\partial L/\partial y$, as explained in the text.

We currently explain this at the end of Section II.A on page 3, starting “In supervised learning, we are interested in adjusting the trainable parameters θ . . .” and we extended the explanation in the revised manuscript.

Action taken: We clarified that the required measurements can be performed in parallel making quantum equilibrium propagation more efficient than the parameter-shift method. For convenience, we replicate the relevant paragraph below. Changes are typeset in green.

However, Onsager reciprocity, Eq. (3), tells us that we can also access the susceptibility $\chi_{\ell j}$ by performing an alternative, much more efficient experiment that instead reveals $\chi_{j\ell}$: apply a small force ν acting on the outputs and observe its influence on the expectation values of the degrees of freedom $\langle \hat{A}_k \rangle$ connected to the trainable parameters: $\chi_{j\ell}(\lambda) = \partial \langle \hat{A}_j \rangle / \partial \nu_\ell$, see Fig. 1 b.

The required measurements of $\langle \hat{A}_j \rangle$ (for any j) under application of a given force component ν_l (at fixed l) can be performed in parallel, in a single experiment. Thus, this approach would already seem to require a number of experiments that scales with the number of outputs N_{out} (number of choices ℓ), which is typically much smaller than the number of trainable parameters. This already would offer a substantial speedup compared with the naive parameter shift method. However, by evaluating explicitly the desired gradient of the loss function, it becomes apparent that only a single experiment is in fact needed: in this experiment, a force vector $\varepsilon = \partial \mathcal{L} / \partial y$, the so-called error signal, is applied to the output degrees of freedom.

4. The last paragraph of Sec IIA says “Onsager reciprocity teaches us how to translate the classical equilibrium propagation approach to quantum devices.” It implies that one definitely needs “Onsager reciprocity” to realize quantum equilibrium propagation. Please clarify.

We thank the referee for the question. Indeed, the validity of Onsager reciprocity, i.e. the symmetry in the susceptibility matrix, is a prerequisite for the applicability of quantum equilibrium propagation. However, we emphasize this is not a narrow prerequisite: Onsager reciprocity in the form needed here actually holds for all quantum systems in equilibrium. For instance, Onsager reciprocity can easily be derived for pure states directly from the Schrödinger equation and it also holds for thermal states.

Action taken: The sentence highlighted by the referee was actually just a summary of the previous paragraph in which we discussed how realising that Onsager reciprocity is at the heart of equilibrium propagation lets us formulate an equivalent method for quantum systems. We clarified the paragraph as follows. Changes are highlighted in green.

“In this way, Onsager reciprocity teaches us how to translate the classical equilibrium propagation approach to quantum devices. Since Onsager reciprocity holds generally in equilibrium systems (SI), quantum Hamiltonians of arbitrary structure can be considered.”

We also added the section “Derivation of Onsager reciprocity and the gradient formula” to the Supplementary Information. For convenience, we replicate the new section below.

Derivation of Onsager reciprocity and the gradient formula

Onsager reciprocity generally holds in equilibrium quantum and classical systems. Here, we recall how its version for the static linear response can be derived for non-degenerate pure eigenstates of closed quantum systems. First, we take the derivative, w.r.t. λ_j , of the expectation value

$\langle \hat{A}_\ell \rangle \equiv \langle \Psi_n(\lambda) | \hat{A}_\ell | \Psi_n(\lambda) \rangle$ computed from the n -th eigenstate $|\Psi_n(\lambda)\rangle$ of the Hamiltonian

$$\partial_{\lambda_j} \langle \hat{A}_\ell \rangle = (\partial_{\lambda_j} \langle \Psi_n(\lambda) |) \hat{A}_\ell | \Psi_n(\lambda) \rangle + \langle \Psi_n(\lambda) | \hat{A}_\ell \partial_{\lambda_j} | \Psi_n(\lambda) \rangle = 2\text{Re} \left\{ (\partial_{\lambda_j} \langle \Psi_n(\lambda) |) \hat{A}_\ell | \Psi_n(\lambda) \rangle \right\} \quad (1)$$

in which we used that $\hat{A}_j = \hat{A}_j^\dagger$. The above expression involves derivatives of the eigenstate $|\Psi_n(\lambda)\rangle$. Projecting the derivative into the eigenbasis, we obtain

$$\begin{aligned} \partial_{\lambda_j} | \Psi_n(\lambda) \rangle &= \sum_m |\Psi_m(\lambda)\rangle \langle \Psi_m(\lambda) | \partial_{\lambda_j} | \Psi_n(\lambda) \rangle \\ &= \sum_{m \neq n} \frac{|\Psi_m(\lambda)\rangle \langle \Psi_m(\lambda) | \hat{A}_j | \Psi_n(\lambda) \rangle}{E_n(\lambda) - E_m(\lambda)} + | \Psi_n(\lambda) \rangle \langle \Psi_n(\lambda) | \partial_{\lambda_j} | \Psi_n(\lambda) \rangle. \end{aligned} \quad (2)$$

The first term on the right-hand side stems from first order perturbation theory, the second term accounts for gauge freedom, i.e., $|\Psi_n(\lambda)\rangle$ could depend on an arbitrary phase which itself depends on λ . Note that the last term is fully imaginary since

$$0 = \partial_{\lambda_j} \langle \Psi_n(\lambda) | \Psi_n(\lambda) \rangle = \langle \partial_{\lambda_j} \Psi_n(\lambda) | \Psi_n(\lambda) \rangle + \langle \Psi_n(\lambda) | \partial_{\lambda_j} \Psi_n(\lambda) \rangle \quad (3)$$

which implies $\langle \partial_{\lambda_j} \Psi_n(\lambda) | \Psi_n(\lambda) \rangle = -[\langle \partial_{\lambda_j} \Psi_n(\lambda) | \Psi_n(\lambda) \rangle]^*$, so $\text{Re} \langle \partial_{\lambda_j} \Psi_n(\lambda) | \Psi_n(\lambda) \rangle = 0$. Inserting the above expression into Eq. (1), we find

$$\partial_{\lambda_j} \langle \hat{A}_\ell \rangle = 2\text{Re} \left\{ \sum_{m \neq n} \frac{\langle \Psi_n(\lambda) | \hat{A}_j | \Psi_m(\lambda) \rangle \langle \Psi_m(\lambda) | \hat{A}_\ell | \Psi_n(\lambda) \rangle}{E_n(\lambda) - E_m(\lambda)} \right\}. \quad (4)$$

Since this expression is symmetric under changing j and ℓ , this results in Onsager reciprocity for the static linear response

$$\partial_{\lambda_j} \langle \hat{A}_\ell \rangle = \partial_{\lambda_\ell} \langle \hat{A}_j \rangle. \quad (5)$$

5. Under the SecIIB, right hand column of page 3, the ‘nudged phase’ requires an or many additional Hamiltonian terms composed of $\sum_l \nu_l A_l$. Also in the example case on page 5, the authors mentioned that “Inspecting the solution also reveals a surprise: the sensor-system couplings are not weak...” These two facts bother us a lot. This is beyond our physics understandings. As the main idea here is to use the ground state equilibrium propagation, such strong sensor-system coupling/s would definitely change the original quantum system ground state. What is/are main condition/s to make it work?

We thank the referee for the good question. We will now answer the two separate points raised by the referee in turn. First, the number of ‘nudge’ terms in the Hamiltonian is simply the number of output operators. Experimentally, this means the experimentalist has to be able to apply a force (e.g. magnetic field, laser drive, etc.) to each of the output operators (whose number, by the way, is typically far smaller than the number of trainable parameters).

The strength of the nudging should ideally be infinitesimal, to obtain the true mathematical training gradient, but we show in the main text that even finite nudge strengths can give very reliable results (see e.g. Figure 2). This observation is important to be able to reduce the number of measurements, since large nudge strength allows to more easily surpass the measurement shot noise.

The second part of the referee’s question refers to the system-sensor couplings in the particular example of supervised training. These couplings are unrelated to the nudge terms mentioned above, instead they are part of the trainable parameters. We understand that the referee is somewhat worried about our observation that system-sensor couplings after training in this particular example turned out not to be weak in comparison to intra-system couplings. This runs counter to the intuition that a sensor should rather couple weakly to the system it tries to sense.

Action taken: In response to the referee’s very valuable question, we revised the training reported in Fig. 2. In the new training runs, we added a regularization for the coupling strength, imposing a ‘soft cutoff’ to keep the coupling small. This actually further improved the results, especially the generalization properties (when training on small patches instead of on the entire phase diagram). Intuitively, this could be the case because the weaker system-sensor coupling leads to a smaller perturbation of the ground state.

We updated Fig. 2 (panels c,f, and g) with our new numerical results and described the new refined strategy in the text. Furthermore, we added a new section to the Supplementary Material with an in-depth discussion of the influence of the system-sensor coupling strength. In summary, while the supervised training task is solved well in both cases, for weak and large system-sensor couplings, the generalizability is better for weak couplings.

For convenience, we reproduce the updated paragraph in the main text as well as the new figure. Changes appear in **green**.

“Ideally, the couplings inside the system of interest, \hat{H}_0 , should be stronger than the couplings to the sensor and within that sensor. This will ensure that the system of interest is only weakly perturbed, while the recognition model can still react strongly to the features of the ground state $|\psi_0\rangle$ of \hat{H}_0 . **To ensure that the system-sensor couplings remain small in our numerical experiment (below), we enforce a soft cutoff on the couplings during the training (SI). . .** We also confirm that the quantum nature of the coupling is important. This can be ascertained by comparing to a sensor that is only allowed to couple to \hat{Z} operators in the chain, which performs much more poorly (Fig. 2 **g**). By contrast, allowing coupling to non-commuting observables yields the observed good accuracy. **We also investigate the influence of the coupling strength between system and sensor (SI). With the soft cutoff we enforced to keep the couplings small, the system learns to predict the correct phases even when it is only trained on patches within each phase. This ability to generalize is less pronounced without the cutoff on the couplings since, in this case, the system-sensor coupling generally become large during training which disturbs the system spins so that their state substantially deviates from the unperturbed state.**”

Figure 2. Supervised learning: Learning recognition of quantum phases using QEP. [...]

We also provide the new section of the SI below:

Phase sensing: analysis of sensor-system coupling effects

FIG. 1: Phase recognition: Influence of the sensor on the quantum many-body system. We show the spatial profile of the expectation values of spin operators in the cluster Ising model chain of length 8 ($\langle \hat{X}_j \rangle$, $\langle \hat{Y}_j \rangle$, and $\langle \hat{Z}_j \rangle$). In the absence of coupling to the phase sensor quantum system, all profiles would be translationally invariant, i.e. constant (zero for Y and Z in this case). The sensor, whose location along the chain is indicated in the first panel, leads to deviations in its vicinity. These deviations depend on the strength of the sensor-system coupling, which can be suppressed using a cutoff during training. Panel marked 'sensor': probabilities of measuring the two qubits of the sensor in one of the three configurations that are trained to indicate the phase. Both with and without cutoff, the correct phase is indicated. Triangle on the right: for the present figure, the parameters of the Ising model were chosen in the vicinity of a phase boundary, in a region where the effect of the sensor on the quantum many-body system is comparatively strong (blue dot), to illustrate the successful suppression of the effect upon introduction of a coupling cutoff.

One of the possible applications of Quantum Equilibrium Propagation (QEP) we proposed in the main text consists in phase recognition. In the experimental implementation, this would work by coupling a small quantum system (called the phase sensor) to a larger quantum many-body system, whose phases are to be recognized. During training, the internal couplings of the sensor as well as its couplings to the system will be updated. After training, the goal is to have a sensor that is able to indicate correctly the phase of the quantum many-body system, which can be changed when varying that system's parameters. Phase indication works by measuring some of the sensor degrees of freedom, such that the result represents a label for the phase.

During our initial numerical experiments, we observed that sometimes the system-sensor couplings become quite strong, comparable to couplings within the quantum many-body system

itself. Although the performance of such a sensor can be nearly perfect, this has the side-effect of introducing relatively strong perturbations on the quantum many-body system. In this section, we analyze these perturbations and how they are suppressed when the coupling strength is reduced.

In Fig. 1, we show the spatial profiles of the spin operators inside the cluster Ising chain. In the absence of a sensor, these would all be spatially constant. The coupling to the sensor introduces perturbations in its vicinity. We observe that these deviations are strongest when the quantum many-body system is placed near a phase boundary, as might be expected from the usual behaviour of linear response near such boundaries.

We can suppress these effects, making them essentially insignificant, by introducing a cutoff on the system-sensor couplings during QEP training. If the cutoff is chosen sufficiently small (smaller than the typical couplings inside the system), we find that the effects become greatly reduced. In practice, we found best convergence and best results when the cutoff was implemented in a "soft" fashion. This was achieved by adding a term to the cost function that is of the type $\sum_j \lambda_{\text{cutoff}} \max(0, |\theta_j| - c)$. Here $|\theta_j|$ is one of the system-sensor couplings (we are summing over all of them), c is the value of the cutoff, and λ_{cutoff} represents the weight (i.e. the slope of the cost function term rising linearly as a function of coupling). For implementing a soft cutoff, we took $c = 0.1$ and $\lambda_{\text{max}} = 0.02$, because this led to good convergence and small values of the resulting system-sensor couplings with correspondingly very weak effects of the sensor on the system behaviour, as demonstrated by the figures in this section. This soft cutoff was applied in all our numerical experiments reported in the main text (except for the analysis of gradients vs nudging and shot noise, which does not depend on these details since it is anyway performed in the early stages of training).

Furthermore, we have analyzed how these perturbations of the system due to the sensor depend on the location in the phase diagram. The results are shown in Fig. 1. We see that without cutoff, the perturbations can be large across whole ranges of the diagram.

FIG. 2: Phase recognition: Sensor effects across the phase diagram. We extract the 'spatial variation' in the cluster Ising chain, defined as the difference of maximum and minimum expectation value of a given operator, such as $\max_j \langle \hat{Z}_j \rangle - \min_j \langle \hat{Z}_j \rangle$. In the absence of a sensor coupled to the system, this would be zero, since the chain then is translationally invariant. We evaluate and plot this variation across the whole phase diagram. When the coupling is left unconstrained during QEP training, the variation is very large, especially in the Z spin component (which has a ferromagnetic coupling and can display strong fluctuations). This indicates that there are strong perturbations in the vicinity of the sensor. Imposing a cutoff on the system-sensor coupling reduces the variation to nearly zero everywhere. Leftmost panels: Spatially averaged expectation value across the phase diagram, for reference.

6. One of the weaknesses of the proposed method is the requirement for the quantum system to reach the ground state. As explained in the manuscript, the efficient preparation of the ground state of arbitrary complex quantum many-body Hamiltonians is generally a very difficult computational task. Although the authors certainly discuss this issue in section II.C, it would be beneficial to include some comments in the discussion in section III when stating the applicability of QEP: "In all these cases, QEP can be applied even [...] while other parts of the Hamiltonian are unknown."

We thank the referee for the question. Indeed physical equilibration is an important topic. We agree that any energy-based neuromorphic approach needs to address the question how easily the physical systems relaxes towards equilibrium, and this is true both for the existing widely studied classical equilibrium propagation as well as for our new quantum equilibrium propagation method.

As we already discuss in an extensive section in the main text (at the end of "C. Experimental Requirements"), fortunately, ground state preparation is one of the most intensively researched questions in quantum simulation and quantum computing, due to its importance. Indeed, there is a multitude of schemes ranging from hybrid approaches involving an external classical computer (e.g. relying on variational quantum eigensolvers) to purely autonomous schemes (e.g. coupling to a cold environment, reservoir engineering, or digital quantum cooling). In addition to this existing section on the experimental requirements, we have now also emphasized the aspect of equilibration once more in the Discussion, following the referee's suggestion.

Action taken: First, to avoid possible confusion, we clarified the distinction between computational complexity and the experimental approaches to efficient equilibration by revising the relevant paragraph at the end of the 'experimental requirements' section. New additions appear in green.

“Finally, QEP requires efficient **experimental** means to approach the equilibrium state, i.e., for the zero-temperature limit, the ground state $|\Psi(\lambda)\rangle$ of the Hamiltonian. **Before discussing physical equilibration, we note that mathematically/ computationally the task of** ground state search for arbitrary Hamiltonians can become hard both for the classical case (NP-hard for general local spin glass Hamiltonians) and for the quantum case (QMA-hard, i.e., hard even for quantum computers in some instances). At the same time, recently, it has been shown that there exist local quantum Hamiltonians for which it would be classically hard to obtain the ground state that nevertheless can be reached via thermalization. **Coming back to physical equilibration dynamics,** the situation in quantum EP is analogous to classical EP, in which equilibration needs to be studied on a case-by-case basis. Fortunately, efficient **experimental** ground-state preparation of complex quantum many-body Hamiltonians is one of the most intensively researched questions in quantum simulation and quantum computing, mirroring analogous progress in classical equilibration.[...]”

In addition, we now added a brief comment to the discussion section near the end of our manuscript, which is reproduced below.

“In all of these cases, QEP can be applied even when the Hamiltonian is hard or impossible to simulate classically, since QEP extracts gradients via the physical response. In addition, it can be employed in cases in which only some aspects of the system are well characterized or accessible while other parts of the Hamiltonian are unknown. **An important requirement is being able to reach the ground state experimentally, and we explained various options in the section on experimental requirements.**”

7. The manuscript would benefit from a comparison with the current training methods available for neuromorphic quantum systems. Without such comparisons, we can't really see how efficient the proposed method fairs.

We thank the referee for the valuable suggestion. As we explain further below and also clarify with our new addition to Figure 1 (please see also our reply above), our method is unique in the realm of analogue neuromorphic computing (thus precluding comparisons in this domain) and has far less stringent requirements than the existing quantum machine learning approaches in the digital domain.

Action taken: We have now added the following comparison and explanation as a new section in the Supplementary Material (for space reasons):

Comparison to other quantum machine learning approaches

So far, physics-based training of neuromorphic quantum systems has only been studied in the realm of so-called quantum machine learning. More specifically, we are talking of variational

quantum circuits, which consist of a sequence of gate operations with continuous parameters, e.g. continuous rotation angles. While originally used for variational ground state search, they have also been employed to train quantum neural networks. However, we emphasize that this ‘digital’ quantum machine learning relies on a fully-fledged quantum computer, with fully controllable qubits (and recent indications are that it even needs to be fully fault-tolerant to achieve some quantum advantage, further increasing the requirements). The training technique there is the parameter shift method, with the slight improvement that parameters are not shifted infinitesimally but rather by $\pi/2$, creating better signal to noise ratio in the required measurements. Research of recent years has also indicated a problem with so-called barren plateaus, where the cost function is effectively flat in wide regions of parameter space, though this can be avoided to some extent by various ideas like cleverly constructed local cost functions.

In contrast to quantum machine learning (variational quantum circuits), quantum equilibrium propagation does not require a working quantum computer, only some experimental platform with many tuneable parameters, like an analogue quantum simulator. This drastically reduces the resource requirements and makes QEP training of large-scale experimental quantum many body systems much more likely in the short term. In addition, the QEP physical training method needs only a number of experiments of order 1 to extract the gradient with respect to all N trainable parameters, very much in contrast to variational quantum circuits, for which the number of required experiments scales like N .

Some other, less significant aspects are comparable between the two methods; e.g., the unavoidable projection shot noise implies that in both techniques a given experiment (for a given set of parameter values) needs to be repeated multiple times when a better estimate of the expectation values is desired.

To conclude, there are many technical uncertainties in the present manuscript that have to be fixed. The main motivation, introduction and discussion are not clear. In fact, they are very ad-hoc and disjointed with the main results. Hence, we can’t approve its publication at Nature Communications.

We thank the joint referees one and two for their many valuable suggestions. In response to their comments, we added new material and explanations which we believe make the manuscript more accessible. In addition, the new simulations we performed in response to their comments further underpin the strength of the QEP method so that overall we feel the manuscript has significantly improved thanks to the referees’ feedback.

Reviewer #3 (Remarks to the Author):

In this manuscript, the authors demonstrate a direct connection between equilibrium propagation — training method for neural networks that focuses on achieving and maintaining equilibrium states (rather than relying solely on gradient-based optimization) — and Onsager reciprocity which asserts a relationship between different thermodynamic forces and fluxes in a system close to equilibrium.

The authors then use this revealed — and I will also add interesting — connection to derive a quantum version of equilibrium propagation which they then test of specific one-dimensional quantum Ising model in the context of both supervised and unsupervised learning and show that their method may be successfully used for phase detection, which is one the main proposed uses of neural network in the context of quantum many body systems.

The paper is skillfully written and was made very clear to understand. The derivation, results and methods are explained clearly and as far as I could tell are sound.

We thank the referee for highlighting our work as interesting, skillfully written and clear.

The method, that was indeed demonstrated to work, if it does indeed work will undoubtedly become a useful tool at the interface of quantum systems and machine learning. I have but one “major” quibble which is – and this is true for all studies presenting methods of this nature – results are always almost exclusively presented on one-dimensional systems which are known to be much easier to solve than two- and more-dimensional systems. As the authors I am sure are well aware, it is the high-dimensional quantum systems that are the real goal of phase detection and yet it is unclear whether the presented method is expected to perform well on scientifically interesting models. If so, then no doubt that the proposed method deserves to be published in a high impact journal such as Nature Communications. The reader will surely appreciate a discussion or even seeing results pertaining to the performance of the proposed method on more challenging quantum many-body models.

We thank the referee for the important suggestion to discuss the application towards more complex quantum many-body systems, which we implemented in the revised version of our manuscript.

To avoid any misunderstanding, we would like to emphasize that our approach is distinct from all the existing other applications of machine learning to quantum many-body systems. Other applications can be grouped into three categories: (i) using training data entirely from computer simulations; (ii) using training data coming from experiments, fed into a classical digital neural network; (iii) envisaged future use of a full-fledged digital quantum computer for quantum machine learning. By contrast, our method is a blueprint to achieve similar goals (detecting quantum many-body phases etc.) directly in an experiment on a quantum simulator platform where this platform is also trained to do the information processing. To our knowledge no comparable method has been proposed. We also refer to our reply to reviewer 1 and the revised Fig. 1 with an overview of the landscape of neuromorphic approaches and where our new scheme fits in.

Importantly, the QEP method can, in principle, be applied to any Hamiltonian of any dimensionality independent of system size. Of course, in the present manuscript we had to illustrate the approach by way of numerical simulations, and so these are limited due to the exponential explosion of the Hilbert space. However, in a future real application, QEP would be performed directly in the experiment, eliminating this challenge.

That being said, to highlight that QEP in its future experimental implementation can, of course, be applied to more complicated models. In the revised manuscript, we now investigate the Kitaev honeycomb model as an example of an interacting model in two dimensions. As explained above, numerical simulations are limited, but we are able to approximate the behavior of a future large-scale experiment by studying modest finite-size systems. For that model, we use QEP to perform phase detection as well as phase exploration in the presence of an additional magnetic field which makes the scenario more complex and interesting by breaking the integrability of the model (please see the changes listed below).

Action taken: We added a new paragraph to the main text which contains the discussion of the results for the Kitaev honeycomb model and we modified Fig. 3 to show the new results. We added the following paragraph in the main text when we discuss the application of QEP to supervised learning for phase sensors and added the new Figure 3.

... Next, we demonstrate our method’s generality by applying it to a more complex model. This is particularly relevant since QEP is a physics-based training technique which in actual future experimental implementations would not be hampered by the exponential growth of the Hilbert space, unlike classical simulations. Specifically, we numerically simulate the training of a phase sensor coupled to a strongly correlated two-dimensional spin model, namely the honeycomb Kitaev model [55]

$$\hat{H}_{\text{Kitaev}} = \sum_{\alpha} \sum_{\mathbf{r}} J_{\alpha} \hat{\alpha}_{A,\mathbf{r}} \hat{\alpha}_{B,\mathbf{r}+\mathbf{r}_{\alpha}} \quad (6)$$

with $\alpha \in \{X, Y, Z\}$ and \mathbf{r}_{α} defined according to Fig. 3 a. The model has three Abelian phases for $J_X \geq J_Y + J_Z$, $J_Y \geq J_Z + J_X$, or $J_Z \geq J_X + J_Y$, respectively, with other parameter values belonging to a non-Abelian phase. We train sensor spins coupled to \hat{H}_{Kitaev} (Methods) to distinguish the Abelian from the non-Abelian phases. Fig. 3 b shows the single-shot accuracy during the training. The best performing phase sensor achieved a single-shot accuracy of 89.4% and a many-queries accuracy of 100% on the evaluation set shown as inset of Fig. 3 b (Methods). The corresponding phase diagram with the predicted phases is depicted in Fig. 3 c.

Phase sensor for the honeycomb Kitaev model. **a** In the Kitaev honeycomb model, spins are coupled with XX , YY and ZZ couplings on a hexagonal lattice. We couple a sensor (two spins) that will be trained to indicate the phase of the ground state. We simulate an 8×2 system with periodic boundary conditions (dashed lines; Methods). **b** Single-shot accuracy on the evaluation set (inset) during the training. **c** Phase predicted by the best-performing sensor (blue: Abelian phase, red: non-Abelian). Following convention, $J_X + J_Y + J_Z = 1$ in this diagram.

We also use QEP to explore the phase diagram of the honeycomb Kitaev model. We added the following paragraph to the section on unsupervised learning and updated Fig. 4 accordingly.

We also consider phase exploration for the honeycomb Kitaev model with an additional magnetic field in $[1, 1, 1]$ direction breaking integrability

$$\hat{H} = \hat{H}_{\text{Kitaev}} + h \sum_{\beta} \sum_{\mathbf{r}} (\hat{X}_{\beta, \mathbf{r}} + \hat{Y}_{\beta, \mathbf{r}} + \hat{Z}_{\beta, \mathbf{r}}). \quad (7)$$

We use QEP to explore how the expectation value of the flux operator $y \equiv \langle \hat{W}_1 \rangle = \hat{X}_1 \hat{Y}_2 \hat{Z}_3 \hat{X}_4 \hat{Y}_5 \hat{Z}_6$, Fig. 4 **d**, changes. While for $h = 0$, $\langle \hat{W}_1 \rangle = 1$ in the ground-state sector, this is no longer the case for $h \neq 0$ (see Methods). One can apply QEP to find the J_{α} that maximize $\langle \hat{W}_1 \rangle$ (Fig. 4 **e,f**), and in an experiment this could be used for systems of any size, including arbitrary additional terms in the Hamiltonian.

Unsupervised learning applications: phase exploration and sensitivity optimization for sensing. **a** Sketch of phase diagram exploration using QEP. An expectation value of interest, $\langle \hat{A}_\ell \rangle$, is maximised. **b** Two example trajectories in the phase diagram of the cluster Ising Hamiltonian (6), each starting at the position in the phase diagram marked with \times and **c** the corresponding loss functions. The colours in **b** show $\langle \hat{X}_0 \hat{X}_4 \rangle$. **d-f** Phase exploration with QEP applied to the honeycomb Kitaev model in the presence of a magnetic field, Fig. 3 **a**. **d** Flux operator \hat{W}_1 defined for the first plaquette. Without magnetic field, $\langle \hat{W}_1 \rangle = 1$. We use QEP to find the parameters J_α for which $\langle \hat{W}_1 \rangle$ is maximal. **e** Trajectories obtained with QEP and different initial conditions, converging to the same point ($\nu = 10^{-2}$). **f** Trajectories for different values of the nudging parameter ν ; larger values decrease the accuracy of the gradient, eventually preventing convergence. **g** Sketch of the concept of sensitivity optimization. The derivative of a an observable of interest, $\langle \hat{A}_\ell \rangle$, w.r.t. a certain parameter (or w.r.t. a vector of parameters) is optimised. Concretely, this scheme can be used to devise optimal sensors or to find phase boundaries in a phase diagram. **h-j** Optimization of $\partial \langle \hat{X}_0 \hat{X}_4 \rangle / \partial g_X$ (magnetic field sensor). **h** Two example runs, each showing the trajectories of two points in the phase diagram of the cluster Ising Hamiltonian (6) and **i** the corresponding loss functions. The two points are initialized at the position marked with \times and converge towards the phase boundary where the derivative w.r.t. g_X is maximal. In the second run, **j** the trajectories first converge towards a phase boundary and then follow it, suggesting that the technique may be employed to trace out phase boundaries. The plots on the right show cuts through the phase diagram at various steps during the training.

We also added the following two sections to the Methods.

Technical details of the phase sensor for the Kitaev honeycomb model

We simulate a lattice of 8×2 spins which is coupled to two additional sensor spins, Fig. 3 a, i.e., 8 plaquettes, under periodic boundary conditions. Since we perform exact diagonalisation to compute expectation values in the ground state, we choose a system size that allows to perform the calculations within a reasonable time frame. We note that the envisioned implementation of the system with a quantum simulator would not have the same limitations. We choose this particular 8×2 lattice to keep the impact of finite size effects as small as possible. In particular, without the correct boundary conditions, contrary to what is expected from the exact solution in the thermodynamic limit $\langle \hat{W}_1 \rangle \neq 1$ at finite system size. This is because the quantization of $\langle \hat{W}_1 \rangle$ is based on Lieb's theorem which only applies in the limit of an infinite system or with the correct, twisted periodic boundary conditions at finite size [58]. Limiting the total number of system spins to 16, we chose to simulate a system of 8×2 spins for which the normal periodic boundary conditions correspond to the twisted boundary conditions needed so that Lieb's theorem holds and $\langle \hat{W}_1 \rangle$ is quantised at finite system size.

We consider couplings between the two sensor spins (all possible combinations of Paulis) as well as couplings to the system spins of the Kitaev model via the operators $\hat{X}_{A,r} \hat{X}_{B,r+r_X}$, $\hat{Y}_{A,r} \hat{Y}_{B,r+r_Y}$, $\hat{Z}_{A,r} \hat{Z}_{B,r+r_Z}$ of one plaquette, Fig. 3 a. See the Supplementary Material for further details about the numerical implementation.

Further details about phase exploration with the Kitaev honeycomb model

In the limit $h = 0$, the Kitaev honeycomb model, Eq. (8) can be mapped to a bi-linear Majorana fermion model and is exactly solvable [55]. In that case, the flux operator defined on one plaquette, e.g. $\hat{W}_1 = \hat{X}_1 \hat{Y}_2 \hat{Z}_3 \hat{X}_4 \hat{Y}_5 \hat{Z}_6$, has eigenvalues ± 1 with the $+1$ eigenvalue corresponding to the ground state sector in which $\langle \hat{W}_1 \rangle = 1$. Switching on the magnetic field $h \neq 0$, the model is not solvable and $\langle \hat{W}_1 \rangle$ is no longer quantized, so we use QEP in the main text to explore how $\langle \hat{W}_1 \rangle$ changes as a function of J_α ; in particular, we fix $h = 0.05 \equiv \text{const.}$ and are looking for the parameters J_α that maximize $y \equiv \langle \hat{W}_1 \rangle$. Note that any other plaquette would yield the same behaviour due to translational symmetry.

Furthermore, we added the sections reproduced below to the Supplementary Information.

Further details about the Kitaev honeycomb model

Implementation of the model

FIG. 3. **Kitaev honeycomb model with the periodic boundary conditions we implemented.** **a** Hexagonal lattice and **b** brickwall representation of the lattice which we used to implement the model.

Fig. 3 **a** and **b** show the system we simulated numerically, including the boundary conditions we employed. The hexagonal lattice can be mapped to a brickwall lattice with the connectivity shown in Fig. 3 **b**.

Phase sensor

FIG. 4. **a** Single-shot accuracy and **b** max-choice accuracy in the phase diagram of the honeycomb Kitaev model.

The training of the phase sensor consists of two steps: a pre-training phase and a second fine-tuning training phase which is shown in Fig. 3 of the main text. During the pre-training phase we train using the adam optimizer for 25 epochs with a learning rate of 0.01, for 5 epochs with a

learning rate of 0.1, for 265 epochs with a learning rate of 0.05 and finally for 173 epochs with a learning rate of 0.01. To suppress the fluctuations in the cost functions in the second fine-tuning training stage, we use the adamax optimizer while keeping the learning rate at 0.01. During both training stages we assume small shot noise with $M = 10,000$. The nudging parameter was set to 0.05. We take the different sizes of the Abelian and non-Abelian phases into account such that during one epoch we sample the same number of points within each phase for one batch. In both training phases, the batch size was set to 6, i.e., 3 points were sampled from each phase, respectively. We show the single-shot accuracy and many-queries ("max choice") accuracy across the phase diagram in Fig. 4 for the best-performing phase sensor. As we can see, the phase sensor for most parts can distinguish the Abelian and non-Abelian phase with a probability close to one and only fails close to the phase boundary.

We thank the reviewer for their positive review and their valuable suggestion. The addition of these new simulation results further substantiates the strength of QEP and we feel that the revisions have significantly improved the manuscript.

Report on NCOMMS-24-39349 titled
“Quantum Equilibrium Propagation for efficient training of quantum systems based on Onsager reciprocity”

In this paper, the authors propose a method for training neuromorphic quantum systems. The method, called quantum equilibrium propagation (QEP), is a quantum version of the widely studied equilibrium propagation (EP) technique that exploits Onsager reciprocity to apply the training procedure to quantum Hamiltonians. The feasibility and applicability of the QEP approach are demonstrated via numerical simulations by solving small-scale supervised and unsupervised learning tasks in many-body quantum physics and sensing.

In our opinion, the current manuscript at the present stage does not have enough merit to be accepted in a prestigious journal such as Nature Communications. To be fair to the authors, we provide detailed questions, comments, and doubts that should be cleared for future submission. Our questions and comments are listed in the order of appearance in the manuscript.

1. Under the introduction and the Discussion, the authors mentioned multiple times about the recent popularity of neuromorphic computing. But, those reasoning does not go very well with what the authors presented in the main text. In that sense, the manuscript is not clearly written. What is the main selling point here? Do the authors like to aid in solving some or a part of the **classical** neuromorphic computing or the **quantum** neuromorphic computing? If the focus was in “quantum neuromorphic one”, how the proposed method is helpful in that system architecture? Is it part of the quantum memristor circuit?
2. The numerical demonstrations are performed for very small systems. Do the authors have any comments on the scalability of the method for large systems? For example, regarding the number of measurements needed or the influence of training samples.
3. In addition to the point 2 above, we do not think that the cost estimate of the training parameters based on the Onsager reciprocity is of the order 1 as suggested by the authors under the figure 1 caption as well as Section II Results. To be specific, let us look at Eq.(3), the Onsager reciprocity. It simply means that the susceptibility matrix χ_{lj} is symmetric. To get the gradient for all the parameters, as we understand based on Eq.(3), instead of finding gradients for n^2 times, we would require $n(n + 1)/2$, assuming the square matrix with the size $n \times n$. Hence, the claim of order 1 is not clear to us and we believe it is not correct.
4. The last paragraph of Sec IIA says “Onsager reciprocity teaches us how to translate the classical equilibrium propagation approach to quantum devices.” It implies that one definitely needs “Onsager reciprocity” to realize quantum equilibrium propagation. Please clarify.
5. Under the SecIIB, right hand column of page 3, the ‘nudged phase’ requires an or many additional Hamiltonian terms composed of $\sum_l v_l A_l$. Also in the example case on page 5, the authors mentioned that “Inspecting the solution also reveals a surprise: the sensor-system couplings are not weak...” These two facts bother us a lot. This is beyond our physics understandings. As the main idea here is to use the ground state equilibrium propagation, such strong sensor-system coupling/s would definitely change the original quantum system ground state. What is/are main condition/s to make it work?
6. One of the weaknesses of the proposed method is the requirement for the quantum system to reach the ground state. As explained in the manuscript, the efficient preparation of the ground state of arbitrary complex quantum many-body Hamiltonians is generally a very difficult computational task. Although the authors certainly discuss this issue in section II.C, it would be beneficial to include some comments in the discussion in section III when stating the applicability of QEP: ”In all these cases, QEP can be applied even [...] while other parts of the Hamiltonian are unknown.”
7. The manuscript would benefit from a comparison with the current training methods available for neuromorphic quantum systems. Without such comparisons, we can’t really see how efficient the proposed method fairs.

To conclude, there are many technical uncertainties in the present manuscript that have to be fixed. The main motivation, introduction and discussion are not clear. In fact, they are very ad-hoc and disjointed with the main results. Hence, we can’t approve for its publication at Nature Communications.